

# Quantifying the dust direct radiative effect in the Southwestern United States: findings from multiyear measurements

Alexandra Kuwano[1], Amato Evan[1], Blake Walkowiak[1], and Robert Frouin[1]

[1]Scripps Institution of Oceanography, University of California San Diego, 9500 Gilman Dr, La Jolla, CA 92093

**Correspondence:** Amato Evan (aevan@ucsd.edu)

**Abstract.** Mineral aerosols (i.e., dust) can affect climate and weather by absorbing and scattering shortwave (SW) and long-wave (LW) radiation in the Earth's atmosphere (the direct radiative effect). It is thought that the dust direct radiative effect is sufficiently strong that the presence of dust can significantly alter surface temperatures, static stability of the atmosphere, and the top of the atmosphere energy balance. Yet despite its importance, understanding of this parameter is so poor that, for

example, the sign of the net direct radiative effect at top of the atmosphere is unconstrained, and thus it is unknown if changes in dust over time cool or warm Earth's climate. Here we develop and apply new methods to estimate the SW direct effect via observations of aerosols and radiation made over a three-year period in a desert region of the southwestern United States. We generate region-specific dust optical properties via a novel data set of soil mineralogy from the Airborne Visible/Infrared Imaging Spectrometer (AVIRIS) instrument, which are then used to model the SW and LW dust direct radiative effect. From

the observations and model output we find that the net dust direct radiative effect, on average, is $-6 \pm 1$ and $-2 \pm 1$ W m$^{-2}$ at the surface and top of the atmosphere, respectively. Our results suggest that the magnitude of the SW component is about twice that in the LW, underscoring the importance of quantifying the iron oxide content of dust since these minerals strongly affect dust SW absorbtivity.

## 1  Introduction

Aeolian dust accounts for the majority of aerosol mass in Earth's atmosphere (Gliß et al., 2021) and there are a number of mechanisms by which dust interacts with the Earth's climate system (Kok et al., 2023). For example, dust suspended in Earth's atmosphere directly affects the Earth's energy budget by absorbing and scattering shortwave (SW) radiation, and absorbing, scattering, and emitting longwave (LW) radiation (Sokolik and Toon, 1996). Since dust is an effective ice nucleating particle, it also indirectly affects the Earth's energy budget by altering ice cloud development (DeMott et al., 2003; Sassen et al., 2003;

DeMott et al., 2010; Rosenfeld et al., 2001). Through absorption of SW and LW radiation dust can alter the atmospheric temperature profile and induce semi-direct affects on the Earth's energy budget (Helmert et al., 2007; Johnson et al., 2004) or perpetuate feedbacks within the Earth's climate system (Helmert et al., 2007; Kok et al., 2018).

Here we focus on improving understanding of the dust direct radiative effect, which is the difference between the net flux in clear-sky (cloud free and dusty) and pristine-sky (cloud and dust free) conditions. In the SW dust typically cools the Earth's

surface and top of the atmosphere while in the LW dust typically induces a warming effect (Liao and Seinfeld, 1998). Di Biagio





et al. (2020) and Kok et al. (2017) separately used observations to constrain model estimates of the globally averaged direct radiative effect of dust at the top of the atmosphere (TOA), both concluding that the sign of this effect could not be constrained based on available data. As such, it is unknown if dust, via the direct effect, warms or cools Earth's climate. One reason for this uncertainty is that dust microphysical properties vary greatly as a function of space and time, so that modeling dust single
scatter properties is highly complex and uncertain (Di Biagio et al., 2019, 2020; Kok et al., 2017; Song et al., 2022).

For example, dust absorptivity is strongly dependent on the concentration of iron oxides in the soils from which the aerosols were emitted (Di Biagio et al., 2019), making this property strongly dependent on source region. Furthermore, there are lingering uncertainties regarding how the dust size distribution evolves over time, with observational studies showing much larger dust particles far from source regions than what is predicted by models (Ryder et al., 2018, 2019; van der Does et al., 2018).
Such an underestimation of the size distribution can result in an overestimation of the SW cooling by dust (Kok et al., 2017). Additionally, although dust particles are highly aspherical, dust single scatter properties are typically calculated assuming spherical particles, and neglecting those shape characteristics can impact the resultant estimates of its radiative effects (Huang et al., 2023).

Given the challenges associated with modeling the the radiative effects of dust, a number of studies have used observations
of radiative fluxes and retrievals of dust physical properties to obtain observation based estimates of dust direct effect in the SW (e.g., Hsu et al., 2000; Di Biagio et al., 2009, 2010; Yang et al., 2009; Kuwano and Evan, 2022) and LW (e.g., Brindley, 2007; Brindley and Russell, 2009; Zhang and Christopher, 2003). Since the main challenge with using observations to estimate the clear-sky SW and LW direct radiative effect of dust is that pristine-sky fluxes can rarely, if at all, be measured, observational methods typically estimate the dust forcing efficiency, which is the direct effect normalized by aerosol optical depth. However,
uncertainties associated with instrumental and retrieval errors, knowledge of the vertical structure of dust and atmospheric temperature, and potential correlation between dust and other radiometrically relevant atmospheric constituents, can produce large uncertainties and even biases in the resulting forcing efficiency estimates (Kuwano and Evan, 2022).

In this study we estimate the clear-sky surface, atmospheric, and TOA SW and LW direct radiative effect and forcing efficiency of dust in the Salton Basin, which is a topographic depression in southeastern California where measurements of
dust and the atmosphere have been made since 2019 (e.g., Evan et al., 2022c). First we develop a modified version of an existing observation-only method to estimate the clear-sky dust SW forcing efficiency (Kuwano and Evan, 2022) in order to calculate the dust SW forcing efficiency and direct radiative effect. We then use novel hyperspectral measurements of the surface to constrain dust optical properties, which are then used along with additional aerosol and meteorological measurements to simulate the SW and LW dust direct radiative effect and forcing efficiency. We find agreement in the observational and modeled estimates
of the direct effect in the SW, which increases our confidence in the methods.

The region of interest contains the Salton Sea, which is a large and shallow body of water located within the terminal basin that is rapidly shrinking due to changes in water management practices (San Diego County Water Authority). There is a growing body of research into the impacts of the shrinking sea on dust emission and the subsequent health effects of exposure (Jones and Fleck, 2020; Parajuli and Zender, 2018). As such, there is need to also improve understanding of the radiative effects
of dust there, which shape potential dust feedbacks onto the local weather and climate.





The remaining portion of this paper is structured as follows. In Sect. 2 we describe the field site, relevant instrumentation, dust identification scheme, and surface soil mineralogy. We then describe and validate the radiative transfer and the linear models used in this study (Sect. 3). In Sect. 4 we present and discuss the dust direct radiative effect and forcing efficiencies obtained from only observations (in the SW) and the output from a radiative transfer model (in the SW and LW). In Sect. 5 we compare our estimates of the SW and LW forcing efficiency and direct radiative effect of dust with similar studies from other regions of the globe. In Sect. 6 we summarize the study and provide suggestions for future work.

## 2   Data sets and products

Here we describe the in situ and satellite based data sets used in this study. A listing of the instruments, products, measurements and retrievals, and their uncertainties is in Table 1.

### 2.1   Field site description, instruments, and measurements

The in situ measurements used in this study were generated over the 2020–2022 time period at a field site in southeastern California (33.17° N, 115.86° W, Fig. 1). The field site lies within the Salton Basin, which is a northwest-southeast oriented rift valley that is bounded to the west, north, and south by the Peninsular, Little San Bernadino, and Orocoppio and Chocolate Mountains, respectively. At the sub sea-level center of the Basin is the terminal Salton Sea, a large and shallow ephemeral lake. The region typically receives less than 100 mm of precipitation each year (Stephen and Gorsline, 1975; NCEI). During the summer daytime surface temperatures often reach values greater than 38 C, while winter temperatures are moderately cool and can fall below freezing at night (Ives, 1949; Imperial County Air Pollution Control District, 2018). The morphology of the area includes alluvial fans, sand and sand dunes, dry washes, paleo lakebed, and rock and vegetated surfaces (Imperial Irrigation District, 2016).

In situ measurements presented here were collected over the 2020–2022 time period at a field site that lies approximately 2.5 km west of the Salton Sea's current western shoreline. The site is immediately east of the Anza desert, and is at the northern and southern edges of the Imperial and Coachella Valleys, respectively (Fig. 1b). Measurements from the site acquired during dust storms have been used to elucidate their meteorological aspects (Evan et al., 2022b, 2023). A photo of the field site instrumentation is in Fig. 2.

### 2.1.1   Solar and infrared radiometers

We obtained surface LW upward and downward fluxes from Kipp and Zonen CG4 and CGR4 pyrgeometers (1A in Fig. 2). These radiometers measure broadband LW fluxes in the 4.5–42 $\mu$m range, are outfitted with Pt-100 thermistors to measure instrument body temperature, and have previously reported relative uncertainties of approximately $\pm 3\%$ (Ramana and Ramanathan, 2006). Longwave fluxes are acquired every second and then averaged over 1 minute intervals. We obtained surface SW upward and downward fluxes from two Kipp and Zonen CM21 pyranometers (2A in Fig. 2). These radiometers measure broadband SW fluxes in the 0.3–2.8 $\mu$m range and exhibit small cosine offsets with a typical maximum relative error of 3% at

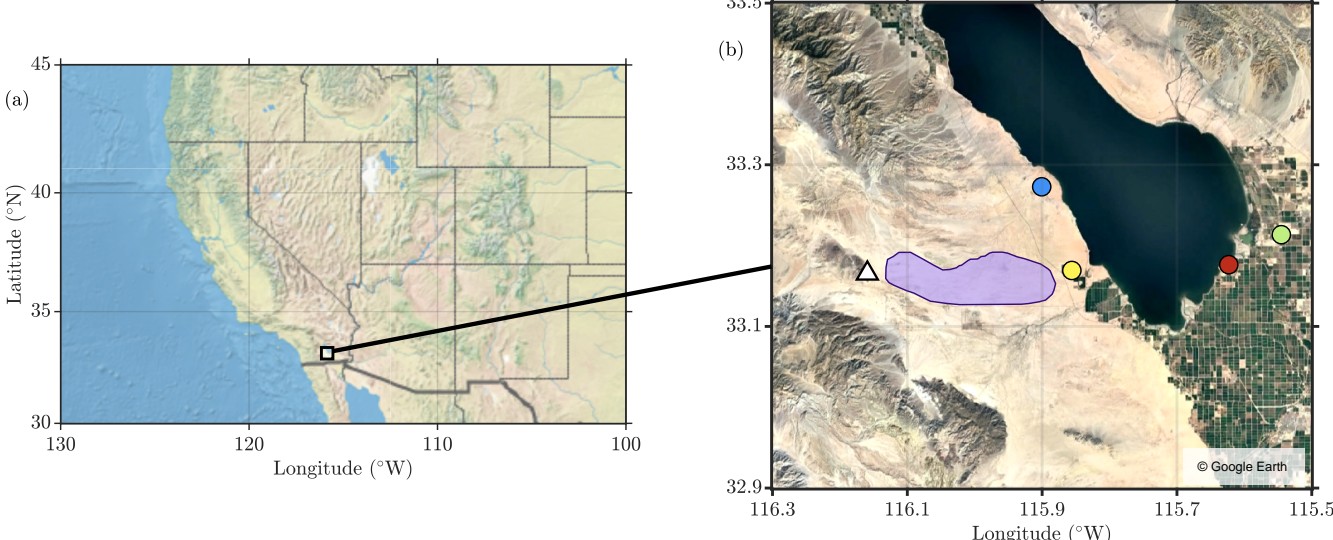

**Figure 1.** Map of the region of interest. Shown is (a) a map indicating the location of the Salton Basin and its relation to the Western United States and (b) key locations surrounding the Salton Sea. The region enclosed by the purple polygon in (b) was used to generate the surface soil mineralogy for the radiative transfer simulations. Also shown in (b) are the locations of the field site (yellow circle), Roundshot camera (white triangle), and the Salton City, Naval Test Base, Sonny Bono, and Niland-English Road TOEM sites (blue, yellow, red and green circles, respectively). The map shown in (a) is created using Natural Earth via MATLAB and (b) is from Google Earth retrieved 26 August 2023.

high solar zenith angle (Zonen, 2004; Ramana and Ramanathan, 2006). Shortwave fluxes are also acquired every second and averaged over 1 minute intervals.

A detailed description of the radiometer calibration is in Appendix B. One of each of the pyranometers and pyrogeometers were factory calibrated on July 2018 and then February 2023, and the pyranometer and pyrogeometer calibration coefficients for these two dates differed by less than 2% and 3%, respectively. We then cross-calibrated the other pyranometer and pyrogeometer during four distinct time periods occurring in 2018, 2021, 2022, and 2023. To do so we placed the instruments side-by-side for time periods ranging from 1–3 weeks and then used linear least square regression to identify the slope of the best-fit line relating the measured voltages, forcing the intercept through zero. For these four time periods the pyranometer and pyrogeometer cross-calibration coefficients differed by up to 2% and 3%, respectively. The factory and cross-calibration coefficients were then applied to the measured voltages by interpolating their values in time, in order to obtain the final LW and SW fluxes. Since the uncertainty in the cross-calibration coefficients were $< 0.3\%$, we assume factory calibration uncertainties of 1% for the pyranometer and 5% for the pyrogeometer as the uncertainty in the radiometer SW and LW fluxes, respectively (Table 1).



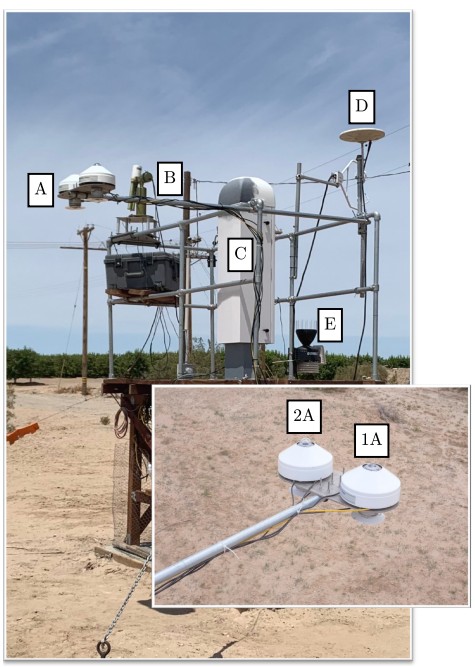

**Figure 2.** Instrumentation at the field site. Shown are radiometers (A), sun photometer (B), ceilometer (C), GPS antennae (D), and meteorology station (E). The inset panel indicates the upward and downward looking pyrgeometers (1A) and the upward and downward looking pyranometers (2A). Radiometer (1A and 2A) image credit: Scott Polach.

### 2.1.2 GPS

We obtained hourly values of total precipitable water $P_W$ retrieved from a Zephyr Geodetic 2 Antenna and Trimble NetR9 GPS receiver (D in Fig. 2), which were processed by SuomiNet (Ware et al., 2000). The relative uncertainty in $P_W$ from the GPS is approximately 10% (Wang et al., 2007; Bevis et al., 1994).

### 2.1.3 Sun photometer

At the field site is a NASA Aerosol Robotic Network (AERONET) CIMEL sun photometer (B in Fig. 2). The CIMEL measures sun collimated direct beam irradiance and directional sky radiance at eight spectral bands centered on 1020, 870, 675, 440, 936, 500, 380, and 340 nm (Holben et al., 1998). Direct solar irradiance measurements are made at 5-min intervals. We use data from the AERONET level 1.5 products processed by the version 3 AERONET algorithm, which provides fully automatic cloud screening and instrument anomaly quality controls in near-real time (Giles et al., 2019). We include dusty observations that were erroneously classified as cloud contaminated using the restoring algorithm described in Evan et al. (2022c). AERONET retrieved aerosol optical depth $\tau$ has a reported absolute uncertainty of 0.01 (an approximate 5% relative error for the field site), and $P_W$ has a reported relative uncertainty of approximately 10% (Holben et al., 1998).





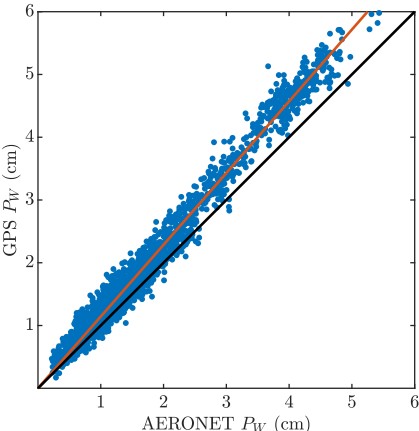

**Figure 3.** A scatter plot of GPS (vertical axis) and AERONET (horizontal axis) retrievals of $P_W$ (blue) made at the field site, the one-to-one line (black), and the linear least-squares regression line, forced through the origin (red).

Since $P_W$ from the GPS is not available at all study times we utilize $P_W$ retrievals from AERONET for this analysis. We calibrated the AERONET $P_W$ based on a comparison with retrievals from GPS. Over the 2020–2022 time period we identified

more than 6,000 simultaneous retrievals of $P_W$ from these two instruments, which are correlated at an r-value of 0.99 (p-value $< 0.01$, Fig. 3). We calibrate the AERONET $P_W$ against that from the GPS to account for a relative negative bias in AERONET $P_W$ by multiplying the AERONET values by the 1.43, which is the slope of the best fit line from a linear regression of the GPS $P_W$ on the AERONET $P_W$, forcing the line through the origin (red line, Fig. 3). The resulting root mean square error in the AERONET $P_W$ is 0.15 cm (10% relative uncertainty).

**2.1.4    Ceilometer**

At the field site is a CL51 Vaisala ceilometer (C in Fig. 2), which is a single lens lidar system that measures attenuated backscatter at a nominal wavelength of 910 nm from the surface to 15 km. The ceilometer generates range corrected backscatter profiles having temporal and vertical resolutions of 36 s and 10 m, respectively. The Vaisala processing software for the CL51 measurements, BLView, identifies clouds based on the vertical gradient of backscatter profiles. Ceilometers have been used to

retrieve the vertical profile of aerosols in the lower atmosphere (Jin et al., 2015; Marcos et al., 2018; Münkel et al., 2007), and the BLView software also retrieves vertical profiles of extinction for the clear-sky atmosphere below heights of 5 km above ground level (AGL). Although details regarding the retrieval process used in BLView are not publicly available, it is possible to approximately reproduce the extinction profile retrievals using typical methods (Fernald, 1984). The CL51 extinction profiles used here have been calibrated to equivalent 500 nm values using the simultaneous retrievals of aerosol optical depth from the

AERONET site (Evan et al., 2022c), such that the AERONET and CL51 retrieved optical depths are identical.

To guard against cloud-contamination in our analysis and results we identify all times where the ceilometer's proprietary software identified a cloud overhead. Since the ceilometer software's cloud detection algorithm often misidentifies thick dust





plumes as clouds, we discard all detected clouds having a base height less than or equal to 2 km AGL (Evan et al., 2022c). We then assume that the sky is at least partially cloudy if a cloud was detected within a 30-minute window by the ceilometer

(Wagner and Kleiss, 2016), and thus do not include these data in our analysis.

### 2.1.5 Meteorological measurements

**Surface meteorology:** A Vantage Pro2 Davis Met Station is also located at the site (E in Fig. 2) and provides measurements of pressure $P$, temperature $T$, specific humidity $q$, and zonal and horizontal wind speed $u$, $v$ at a height of 2 m, and which are logged at a 1 min resolution. These data are separately processed and available through MesoWest at a nominal resolution of

15 min (Horel et al., 2002).

**Radiosondes:** We obtained vertical profiles of temperature $T(z)$, pressure $P(z)$, and specific humidity $q(z)$ from 15 radiosondes launched at the field site during the 2020–2022 time period using a Vaisala sounding system (dates and times listed in Table A1). Since these soundings typically extend to heights of 20–25 km, and in order to conduct the radiative transfer simulations, we extend these profiles to heights of 32 km using radiosondes from the San Diego sounding station (NKX), and

then from 32 to 95 km using a standard mid-latitude summer atmospheric profile (Anderson et al., 1986).

### 2.2 Satellite data

We utilize observations of the TOA SW upward flux and outgoing longwave radiation (OLR), and estimates of the broadband SW surface albedo $\alpha$ from the Clouds and Earth's Radiant Energy System (CERES) Single Scanner Footprint (SSF) level 2 data product (Loeb et al., 2016; Su et al., 2015a, b). CERES is a space-borne instrument that measures TOA radiance in

the SW (0.2–5 $\mu$m), window (8–12 $\mu$m), and total (0.2–100 $\mu$m) spectral intervals at a spatial resolution of approximately 20–25 km. LW radiance is estimated by taking the difference between the total and SW radiances. Instantaneous CERES SSF measurements are collected along-scan of the CERES footprint as it traverses the Earth's surface.

We use daytime CERES SSF level 2 data collected from the NASA Aqua, NASA Terra, National Oceanic and Atmospheric (NOAA) Suomi National Polar-orbiting Partnership (NPP) and NOAA-20 sun-synchronous satellites. The SSF data corre-

sponding to the CERES instruments flying on the NASA satellites include retrievals/products from the Moderate Resolution Imaging Spectroradiometer (MODIS) instruments on those platforms, while those flying on the NOAA satellites include that from Visible Infrared Imaging Radiometer Suite (VIIRS) instruments. This data product also includes footprint-averaged deep blue aerosol optical depth retrievals (Hsu et al., 2013), surface albedo (Rutan et al., 2009), and $P_W$ from the Goddard Earth Observing System Model version 5.4.1 reanalysis (Rienecker et al., 2008).

In order to obtain CERES data representative of the conditions over the field site, we limit our analysis to footprints having a nadir center-point within approximately 25 km of the field site (Fig. 4). We obtain clear-sky fraction from the CERES level 2 clear/layer/overlap condition percent coverage parameter (Wielicki et al., 1996) and further limit our analysis to satellite footprints having clear-sky fraction $\geq 95\%$. We only utilize CERES data if the measurement was generated within 30 min of an AERONET measurement. These constraints resulted in 1,639 CERES level 2 SSF products over the 2020–2022 time period

that were considered for our analysis (Fig. 4).



**Table 1.** Instrumentation and measurements or retrievals used in this study at either the field site (top) or from space-borne platforms (bottom). Uncertainties are indicated for measurements and retrievals that are used to calculate the dust direct radiative effect and forcing efficiency.

| Instrument/Product | Measurement/Retrieval | Uncertainty |
| --- | --- | --- |
| *in situ* | | |
| CIMEL Sunphotometer | $\tau$ at 0.5$\mu$m | 5% (0.01) |
| | Total precipitable water $P_W$ | 10% |
| | Solar zenith angle $\theta$ | |
| | Fine-mode-fraction $f$ | |
| Trimble GPS | Total precipitable water $P_W$ | 10% (1.5 mm) |
| CM21 Pyranometers | Surface SW upward and downward flux | 1% (2–7 Wm$^{-2}$) |
| CG4 Pyrgeometer | Surface LW upward flux | 5% (28 Wm$^{-2}$) |
| CGR4 Pyrgeometer | Surface LW downward flux | 5% (16 Wm$^{-2}$) |
| Davis Met Station | 2 m $T, P, q$ | |
| Vaisala Radiosonde | $T(z), P(z), q(z)$ | |
| *space-borne* | | |
| CERES SSF (L2) | TOA SW upward flux | 2% (4 Wm$^{-2}$) |
| | TOA LW upward flux | 1% (3 Wm$^{-2}$) |
| | Solar zenith angle $\theta$ | |
| | Clear-sky fraction | |
| | Broadband surface albedo $\alpha$ | 10% (0.02) |
| | Total precipitable water $q$ | 10% |

## 2.3 Other data sets

We utilize vertical profiles of temperature and geopotential height over the field site from the Japan Meteorological Agency 55-year reanalysis (JRA-55), which are available at 6-hourly time increments at a 0.5° spatial resolution (Japan Meteorological Agency, Japan, 2013). The JRA-55 reanalysis data is included in this study because this dataset was the only reanalysis available from the National Center of Atmospheric Research (NCAR) with a 4DVar data assimilation scheme and through 2019 to 2022. We use measurements of concentrations of particulate matter having diameter under 10 $\mu$m (PM$_{10}$) from tapered element oscillating microbalance (TEOM) instruments at several monitoring sites located around the periphery of the Salton Sea (filled circles in Fig. 1). These data are only available at a 1-hour temporal resolution from the California Air Resources Board. Lastly, we obtained images of dust storms from a 360° Roundshot web camera that is maintained by the Imperial Irrigation District. The camera lies at an elevation of 300 m and is 28 km west of the field site (filled triangle in Fig. 1). Roundshot images are available at approximate 10 minute intervals during daytime hours via https://iid.roundshot.com/anza-borrego/#/ and are typically unavailable during July–September.





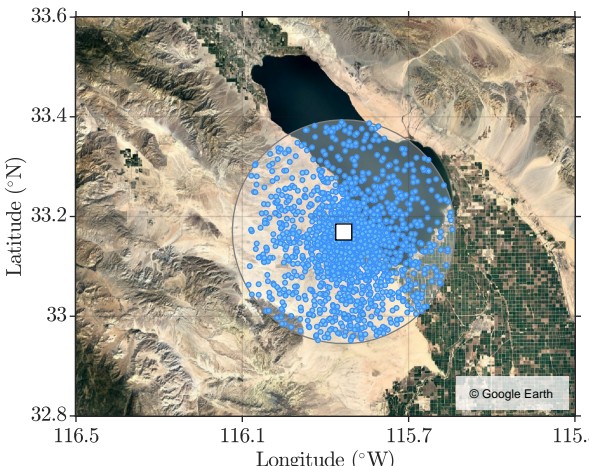

**Figure 4.** Nadir-looking points of the CERES SSF level 2 data considered in this study. Shown are CERES SSF footprints (blue, closed circles) with centers within approximately 25 km of the field site (white square) during daytime and both dusty and non-dusty conditions. CERES SSF footprints shown here are selected when the CERES clear-sky fraction is greater than or equal to 95% (i.e. cloud free). The map shown is from Google Earth retrieved on 26 August 2023.

## 2.4  Identifying dust storms

Here we describe the method employed to identify dust storms passing over the field site during the daytime hours. Firstly, potential dust storms were manually and subjectively identified via visual inspection of backscatter profiles from the site ceilometer for the 2020–2022 time period. We typically selected times of potential dust when there is a strong ($> 3$ a.u.) and persistent (approximately $> 30$ min) backscatter signal near the surface that extends to heights of 500 m to 3 km, which is typical of dust storms over the region (e.g. Evan et al., 2022b, 2023). For example, ceilometer profiles for a 24-hour time period beginning at 02:00 UTC on April 22, 2022 indicate the likely presence of dust during this time period, and thus would be flagged as a potential dust storm for further analysis (Fig. 5a). From these potential storms, we assume individual AERONET measurements are made during dusty scenes if the AERONET fine mode fraction is $< 0.5$ (Evan et al., 2022c) and if $PM_{10}$ from one of the nearby TOEM stations is $\geq 50$ $\mu g$ m$^{-3}$ within an hour of the observation (Hoffmann et al., 2008). We note that if fine mode fraction data is missing we substitute a 440–870 nm Angstrom Exponent threshold of $\leq 0.25$ as a substitute, which is based on a comparison of fine mode fraction and Angstrom Exponent data for other confirmed dust cases (not shown). These additional criteria limit our analysis to clear-sky conditions and daytime data (e.g., mauve boxes in Fig. 5a).

When possible, we additionally confirm the presence of dust in the region via visual inspection of Roundshot camera imagery (location in Fig. 1), specifically identifying dust in the eastward-looking–towards the field site–direction (e.g., Fig. 5b). Since we have observed days when there was dust at the surface but smoke from regional biomass burning events suspended above the dust layer, we excluded dust events if there was an elevated aerosol layer apparent in the ceilometer backscatter profiles or if inspection of visible satellite imagery for these events showed smoke in the region.





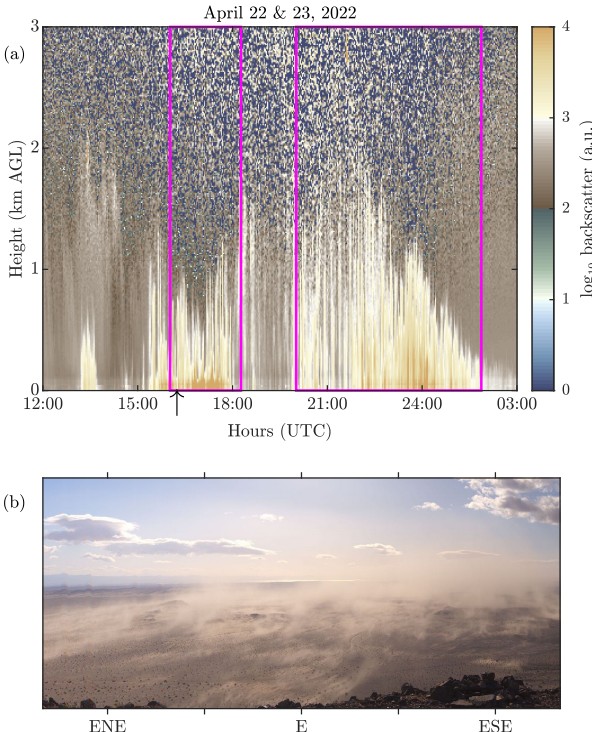

**Figure 5.** (a) Example of a backscatter profile generated from the site ceilometer during a dust storm on 22 and 23 April 2022. Times enclosed by the magenta lines were identified as being dusty after analysis of additional data sources. (b) Eastward looking Roundshot image captured at 16:00 UTC on this date (black arrow on horizontal axis in (a). The Roundshot image in (b) can be obtained via https://iid.roundshot.com/anza-borrego/#/ (accessed 22 September 2022).

This method to identify dust storms resulted in the identification of approximately 2,215 AERONET observations during which there was a dust storm over the field site during 2020–2022. This number is reduced to 1,964 when we also limit the data to times when SW and LW radiometer measurements are available from the field site. We note that there were other dust storms over the site that were not included in this analysis because of intermittent issues with instrumentation (e.g., loss of site

power, contamination of instrument optics, etc.), or because dust storms in the region are oftentimes accompanied by cloud cover (e.g. Evan et al., 2023), as such, this estimate does not represent a complete accounting of all dust storms in the area. Of the 1,639 satellite observations that were collocated to the field site, only 43 were made within 30 minutes of an identified clear-sky dust event, and thus are used in our analysis.

Lastly, most dust events over the field site are associated with strong (e.g., $> 10$ m s$^{-1}$ 10 m wind speeds) westerly winds

(Evan, 2019; Evan et al., 2023). For example, for the April 22, 2022 case the 10 m wind speeds and gust measured at the site were westerly and typically 10 and 20 m s$^{-1}$, respectively (not shown). Furthermore, these dusty air masses typically originate at altitudes above the mountain ridges that lie to the west of the field site, which have heights of 2–3 km, before they





descend along the lee-side mountain slopes, passing over generally uninhabited landscapes. As such, and given the fine mode fraction threshold test required to classify an observation as being acquired during a dust storm, we assume that during dust

storms the aerosols over the site are dominated by mineral species, and that other aerosols constitute a negligible fraction of the total burden. While we have no observations to independently verify this assumption, the agreement between the modeled and observed fluxes during these dust events is indirect evidence of its general validity (Sect. 3.1.3).

## 2.5   Surface soil mineralogy

In order to calculate the complex refractive index of dust over the field site we estimate the average soil mineralogy over the

dust emitting regions that are typically upwind (i.e., to the west) of the field site (purple polygon in Fig. 1), which are identified from analysis of satellite imagery of dust outbreaks and eye-witness accounts. Surface soil mineralogy is from The Airborne Visible/Infrared Imaging Spectrometer - Classic (AVIRIS-C), which is a passive imaging spectrometer that operates in the 0.41–2.45 $\mu$m wavelength range and is designed to operate aboard NASA's Earth Resources 2 aircraft (Chrien et al., 1990). The AVIRIS-C measurements used here were collected over the region in 2018, and from these data the approximate mineral

abundance for the following nine minerals are retrieved: calcite, chlorite, dolomite, goethite, gypsum, hematite, illite, kaolinite, and montmorillonite (Thompson et al., 2020). Also retrieved from AVIRIS-C are estimates of the fractional cover of major land surface types, of which fractional bare soil cover is used here.

## 3   Models

### 3.1   Radiative transfer model

In this study we utilize the SW Rapid Radiative Transfer Model (RRTM) version 2.5 (Atmospheric and Environmental Research, 2004) and LW RRTM version 3.3 (Atmospheric and Environmental Research, 2010) from the Atmospheric and Environmental Research (AER) Inc (Mlawer et al., 1997; Mlawer and Clough, 1997, 1998). This version of RRTM is a band transmission model that evaluates radiative transfer at 14 spectral bands ranging from 0.2–12.2 $\mu$m and 16 spectral bands from 3.1–1000 $\mu$m (Iacono et al., 2008; Clough et al., 2005). The model uses the correlated-$k$ method to treat gaseous absorption. In

the SW code, the correlated-$k$ method is applied to the solar source function, which derives the incoming solar flux at the TOA (Mlawer and Clough, 1998). We use 8 streams in the Discrete Ordinate Radiative Transfer (DISORT) to solve radiative transfer for multiple scattering in both the SW and LW spectrum (Iacono et al., 2008). Each SW band uses a present-day solar source function and a spectrally constant broadband surface SW albedo $\alpha$. Correspondingly each LW band uses a constant broadband LW $\alpha$ of 0.01. We assume a $CO_2$ mixing ratio of 417 ppm, which is the approximate average atmospheric $CO_2$ concentration

measured at Mauna Loa in mid to late 2021. Other gases that are included in the model are water vapor, nitrogen, ozone, nitrous oxide, methane, oxygen, carbon monoxide, and the halocarbons $CCL_4$, CFC112, CFC12, and CFC222. The model assumes Lambertian reflection at the surface. RRTM SW has been extensively validated against the Line-by-line Radiative Transfer Model (LBLRTM), an accurate line-by-line model that is continuously validated against observations; the coefficients used





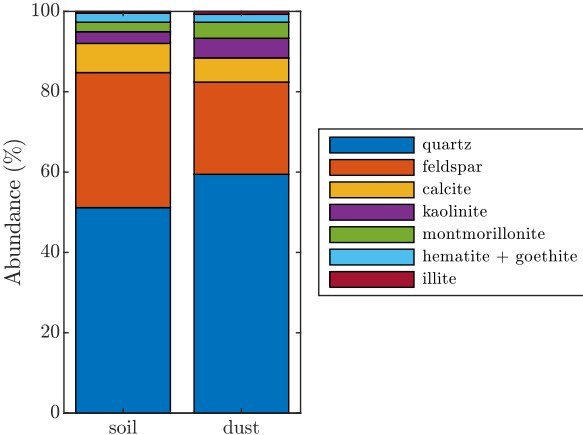

**Figure 6.** Mineral abundances of the surface soil and dust, based on AVIRIS-C retrieved mineral abundance retrievals averaged over the polygon in Fig. 1.

in the correlated-$k$ method are developed via LBLRTM. In the SW, RRTM is in agreement within 1.5 W m$^{-2}$ to LBLRTM

(Clough et al., 2005).

For the RRTM simulations, we estimate the dust complex refractive index (CRI) using the surface soil mineralogy described in Sect. 2.5 and the methods described in Walkowiak (2022). Briefly, we first average the mineral abundances over the polygon shown in Fig. 1, weighing each 20 m horizontal resolution AVIRIS-C grid cell by its corresponding bare soil fraction. The retrieved AVIRIS-C soil mineralogy is 7% calcite, 3% kaolinite, 2% goethite, 2% montmorillonite, and 0.5% hematite and

illite, with the abundances of chlorite, dolomite, and gypsum being $< 0.1\%$ (Fig. 6).

In order to generate a CRI from the AVIRIS-C surface soil mineralogy we first partition the surface mineralogy into clay and silt sizes. To do so we assume that the fractional surface soil abundance of a mineral $m$, given by AVIRIS-C, is

$$m = f_c m_c + f_s m_s \tag{1}$$

where $m_c$ and $m_s$ are the soil mineral abundances in the clay and silt sizes, respectively, and $f_c$ and $f_s$ are the fractional

abundances of clay and silt in the soil, respectively, which are calculated from the soil probability size distribution in Kok (2011). We also define the ratio of mineral abundance $r_m$ in the clay and silt size ranges as $r_m = m_c/m_s$, which allows us to express $m_s$ and $m_c$, via Eq. 1, as

$$m_s = \frac{m}{f_c r_m + fs} \tag{2}$$

and

$$m_c = \frac{m - m_s f_s}{f_c} \tag{3}$$

We obtain $r_m$ for each AVIRIS-C mineral from the clay and silt fractional abundances for the Calcaric Fluvisols soil type in Claquin et al. (1999) and then estimate the fractional contribution of each to the soil clay and silt sizes via Eqs. 2,3. We assume





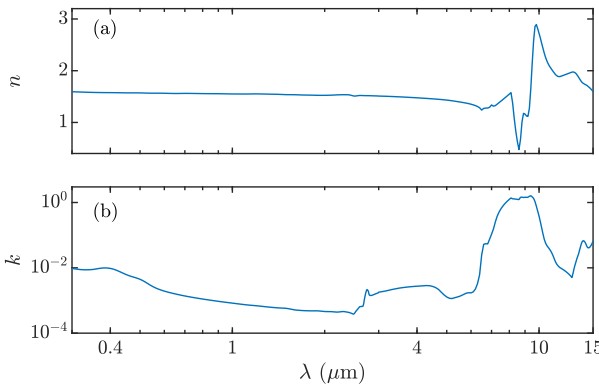

**Figure 7.** (a) Real and (b) imaginary parts of the dust CRI calculated from the AVIRIS-C surface soil mineralogy, over the 0.3–15 $\mu$m spectral range.

that the unclassified abundances in the clay and silt sizes are comprised of quartz and feldspar, two abundant minerals that are not retrieved by AVIRIS-C due to their relatively uniform optical properties across the solar spectrum. We proportionally assign the quartz and feldspar fractions within each soil size class via their relative abundances for the same soil type (Claquin et al., 1999). For this case, the total soil abundances of quartz and feldspar are 51% and 34%, respectively (Fig. 6). We obtained similar results when repeating this process but using the Luvic Yermosols soil type abundances from Claquin et al. (1999), and using the reported abundances for these two soil types via Journet et al. (2014, not shown).

Having partitioned the AVIRIS-C soil mineralogy into the silt and clay sizes, we follow the methods described in Scanza et al. (2015) in order to generate a corresponding dust mineralogy, which slightly differs to that for the surface soil since minerals in the clay sizes are more abundant in the aerosol than in the surface (Fig. 6). We also follow the methods of Scanza et al. (2015) to estimate a resulting CRI from the dust mineralogy. Here, in the solar part of the spectrum we obtain the characteristically flat real part of the refractive index $n$ (Fig. 7a) and an increasing imaginary part of the refractive index $k$ for decreasing wavelength to 0.4 $\mu$m (Fig. 7b), which is due to the abundance of the iron oxides hematite and goethite (Fig. 6) that absorb solar radiation at these wavelengths (e.g. Doner et al., 2019). In the longwave part of the spectrum the peak in $k$ in the 8–10 $\mu$m range is due to strong absorption features in quartz, feldspar, kaolinite, and montmorillonite, and these same minerals also contribute to the trough and peak in $n$ at approximately 8.5 and 10 $\mu$m, respectively.

We note that we obtained a qualitatively similar CRI when using a dust size distribution that includes more coarse dust particles (Meng et al., 2022), and when substituting the Maxwell–Garnett mixing method with the Bruggeman or volume mixing methods (Chỳlek et al., 1988; Bohren and Huffman, 2008).

### 3.1.1 Dust single scatter properties

We obtain dust single scatter properties from the Texas A&M University dust 2020 (TAMUdust2020) version 1.1.0 database of optical properties of irregular aerosol particles (Saito et al., 2021). This database generates single scatter properties of





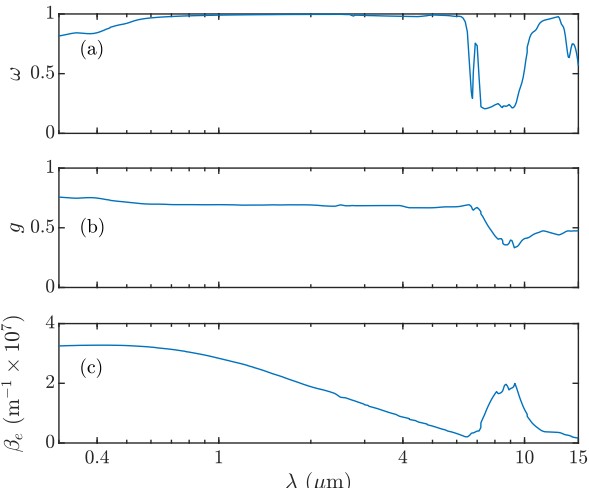

**Figure 8.** Dust single scatter properties used in RRTM and based on the CRI derived from AVIRIS-C (Fig. 7). Plotted are the (a) single scatter albedo, (b) asymmetry parameter, and (c) volume extinction over the 0.3–15 $\mu$m spectral range.

randomly-oriented and irregular-shaped dust particles given a CRI and degree of asphericity by considering ensembles of
at least 20 irregular hexahedral particles. We use the previously described dust CRI (Fig. 7) and use the default model dust asphericity, which is consistent with the global mean dust particle aspect ratio reported in Huang et al. (2020).

We estimate the spectrally-resolved extinction coefficient $\beta_e(\lambda)$, single scatter albedo $\omega(\lambda)$, and asymmetry parameter $g(\lambda)$ from the model output and an estimate of emitted dust size distribution (Meng et al., 2022), using typical methods (e.g. Seinfeld and Pandis, 2016). The resulting single scatter properties exhibit typical characteristics of dust (e.g. Highwood and
Ryder, 2014), including increasing $\omega$ with increasing wavelength (Fig. 8a) and scattering in the forward direction (Fig. 8b), both in the visible, and decreasing $\beta_e$ with increasing wavelength, except for a peak in the 8–10 $\mu$m spectral range (Fig. 8c).

### 3.1.2 Radiative transfer model simulations

We conducted RRTM simulations corresponding to the AERONET observations made during the identified dust storms. For the model simulations we define 107 radiative levels and specify vertical profiles of pressure, temperature, and water vapor from
reanalysis (Sect. 2.3). We calibrated the reanalysis water vapor profiles so that the resulting total precipitable water $P_w$ equaled that retrieved from AERONET. For each simulation we specify the broadband SW albedo given by the site pyranometers, which ranged from 0.23–0.38. We used a constant broadband LW surface albedo of 0.01, which is an average LW surface emissivity retrieved from the CERES observations used in the study (Fig. 4). We estimated the surface temperature for each simulation using the measured upwelling LW flux at the site and the average CERES surface emissivity. We specify the vertical
distribution of dust extinction at 500 nm via the calibrated retrievals from the ceilometer. Spectral variability in extinction is estimated from the extinction coefficient output from TAMUdust2020 (Fig. 8c). We conducted a separate set of simulations with aerosol optical depth set to zero in order to quantify the model dust radiative forcing.





The contribution of fine dust particles to the size distribution increases with increasing height, so that dust single scatter properties are also a function of height. We re-ran a subset of the RRTM simulations with height specific single scatter properties calculated using a theoretical and size-dependent expression for the vertical dust concentration under steady-state conditions (Shao, 2008). We obtained similar fluxes when assuming single scatter properties varied with height as for the simulations when we assumed those properties were independent of height (not shown). We interpret this result as reflecting the relatively low vertical extent of most dust advected over the field site, which is typically below 2 km above ground level (AGL). As such, for simplicity we assume that dust single scatter properties are invariant with height in the RRTM simulations.

Lastly, we also conducted a set of RRTM simulations corresponding to clear-sky and daytime scenes during which we obtained vertical profiles of pressure, temperature, and moisture from 15 radiosondes launched at the site during the 2020–2022 time period on days when dust storms occurred (radiosonde launch times are in Table SA1). These simulations were carried in a manner identical to those described above except that the model was forced with the sounding profiles rather than reanalysis.

### 3.1.3 Radiative transfer model uncertainty

We assess the uncertainty in simulated SW and LW fluxes via comparison to observations. We characterize the model uncertainty as the bias-corrected root mean square error (RMSE) between modeled and measured net fluxes, since biases in the modeled fluxes do not affect our calculations of the direct radiative effect or forcing efficiency. We begin by evaluating the modeled surface fluxes for the simulations using the radiosonde measurements, which include 13 cases for the SW flux comparisons and 15 cases for the LW flux comparison (for two radiosondes the pyranometers were not operational). There was a $-10$ and $-8$ W m$^{-2}$ bias (2% and 5% relative bias) in the modeled net surface SW and LW fluxes, respectively. A comparison of the bias-corrected net SW (Fig. 9a) and LW (Fig. 9b) surface fluxes indicates that the model reasonably reproduces the observed fluxes; the bias-corrected SW and LW RMSE is 6 W m$^{-2}$ (1% relative error) and 4 W m$^{-2}$ (2% relative error), respectively. We note that the differences between the modeled and measured fluxes were not correlated to aerosol optical depth $\tau$, although the small sample size precludes definitive assessment of the impact of dust on model uncertainty. Here we do not compare modeled and satellite-observed TOA fluxes since only two clear-sky satellite measurements were available during the radiosonde launches.

We also estimate the uncertainty in modeled net surface fluxes during dust storms and for times when we do not have measurements from radiosondes, which includes 1,964 and 43 observations at the surface and TOA, respectively. For these cases and in the SW and LW we note good agreement between the modeled and measured fluxes, as evidenced by correlation coefficients of 0.99 and 0.85 for the surface SW and LW flux comparisons (Figs. 10a,b) and 0.97 and 0.93 for the TOA SW and LW flux comparisons (Figs. 10c,d). We note biases in the modeled surface SW and LW fluxes of 4 and 7 W m$^{-2}$, and $-56$ and $-21$ W m$^{-2}$ for the modeled SW and LW fluxes at TOA. The bias-corrected RMSE for the net SW and LW surface fluxes are 17 and 18 W m$^{-2}$, and for the net SW and LW TOA fluxes are 26 and 13 W m$^{-2}$, both respectively.

It is likely that the model flux RMSE for the cases when radiosondes are available are less than that for time periods when radiosondes were not available because of the better representation of the vertical structure of the atmosphere and because we





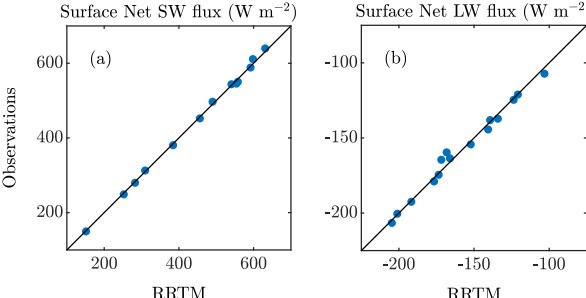

**Figure 9.** Comparison of measured (vertical axes) and bias-corrected modeled (horizontal axes) surface net radiative flux in the (a) SW and (b) LW during times when radiosondes were launched from the field site. The one-to-one line is also shown (black).

corrected any obvious errors in the ceilometer extinction profiles for the times radiosondes were launched. Furthermore, it is also likely that the RMSE and biases for the TOA fluxes are larger than those at the surface because the satellite footprints cover a wide area that can span different surface types and atmospheric conditions (Fig. 4). We note that the errors in the modeled

fluxes were poorly correlated to the accompanying $\tau$, where $r^2$ values remained below 0.02 for all the flux comparisons. The relative RMSE values reported in Fig. 10 are used to estimate error in the RRTM output fluxes in subsequent estimation of the dust direct radiative effect and forcing efficiency.

## 3.2 Linear model and uncertainty

We develop a linear model of net SW flux at the surface $S_0$ and TOA $S_\infty$ in order to generate observational estimates of the

SW dust forcing efficiency and direct radiative effect. To do so we assume that these fluxes can be approximated as linear functions of $\tau$, $P_w$, cosine of the solar zenith angle $\mu$, and surface SW albedo $\alpha$

$$S_0 = \frac{\partial S_0}{\partial \tau}\tau + \frac{\partial S_0}{\partial P_w}P_w + \frac{\partial S_0}{\partial \mu}\mu + \frac{\partial S_0}{\partial \alpha}\alpha + S_0^* \tag{4}$$

and

$$S_\infty = \frac{\partial S_\infty}{\partial \tau}\tau + \frac{\partial S_\infty}{\partial P_w}P_w + \frac{\partial S_\infty}{\partial \mu}\mu + \frac{\partial S_\infty}{\partial \alpha}\alpha + S_\infty^* \tag{5}$$

where the $S^*$ terms are constants representing net fluxes for a pristine and dry atmosphere over a completely absorbing surface with the sun at the horizon, and the $\partial S/\partial$ terms are the sensitivities of the net solar fluxes to the independent variables. These sensitives and constants can be estimated via multivariate linear regression given measurements of $S_0$ or $S_\infty$, $P_w$, $\mu$, and $\alpha$.

In order to justify these linear models of solar fluxes and quantify the uncertainty in each, we estimated the sensitivity terms and $S_*$ in each using simultaneous measurements of $S$, $\tau$, $\alpha$, and $P_w$ from the field site (Eq. 4) and satellites (Eq. 5). These

fluxes are the same data as was used to evaluate the RRTM output (Fig. 10a,c), which include 1,964 observations at the surface and 43 observations at TOA. We then calculated surface and TOA net solar fluxes via these linear models. A comparison of the measured and modeled fluxes suggests that the linear model is able to reproduce much of the variability in the measured fluxes




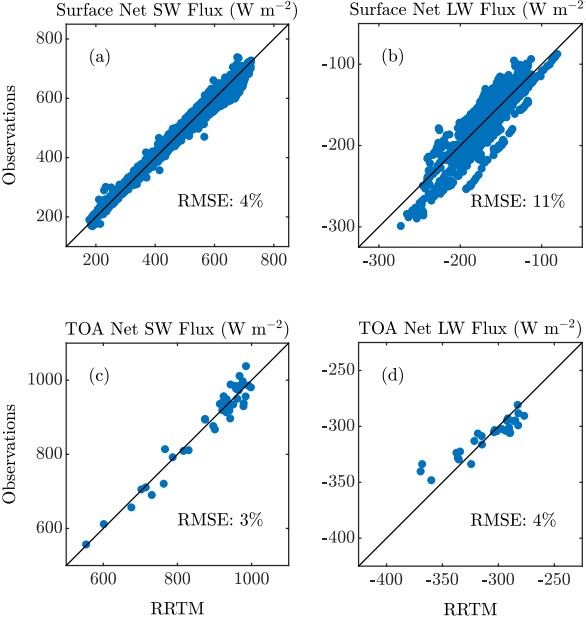

**Figure 10.** Comparison of measured (vertical axes) and bias-corrected modeled (horizontal axes) surface (top row) and TOA (bottom row) net radiative fluxes in the (a,c) SW and (b,d) LW. The one-to-one black line and the relative modeled flux RMSE is also shown in each plot. We note that the axes for each panel are distinct.

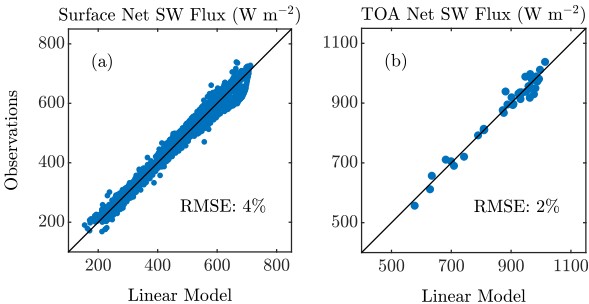

**Figure 11.** Comparison of observed (vertical axes) and linear modeled (horizontal axes) net SW fluxes at the (a) surface and (b) TOA. Modeled fluxes are estimated via multivariate linear regression (Eqs. 4,5). The one-to-one line is shown in each plot (black) as is the relative RMSE.

(Fig. 11). The percent variances of the observations that are explained by the linear models (i.e., $r^2$-value) and their RMSE are 98% and 20 W m$^{-2}$ (4% relative error) at the surface, and 97% and 20 W m$^{-2}$ (2% relative error) at TOA, both respectively.





## 4 Results

In this section we present and discuss the dust direct radiative effect and forcing efficiencies obtained from the linear (in the SW) and radiative transfer (in the SW and LW) models.

### 4.1 Observational estimates of the SW instantaneous direct radiative effect and forcing efficiency

We start by estimating the dust direct radiative effect and forcing efficiency using only observational data.

#### 4.1.1 Observational methodology

This method to estimate the instantaneous dust direct radiative effect and forcing efficiency is a modified version of that described in Kuwano and Evan (2022). Firstly, the clear-sky direct radiative effect of dust $\zeta$ is defined as the difference between the net clear $F$ and pristine-sky $F_p$ fluxes,

$$\zeta = F - F_p \tag{6}$$

In the SW part of the spectrum $F$ can be approximated at the surface or TOA via Eqs. 4,5, and pristine sky fluxes are obtained by setting $\tau = 0$ in each. The resultant SW dust direct radiative effect at the surface $\zeta_0^S$ and at TOA $\zeta_\infty^S$ are then

$$\zeta_0^S = \frac{\partial S_0}{\partial \tau} \tau \tag{7}$$

and

$$\zeta_\infty^S = \frac{\partial S_\infty}{\partial \tau} \tau \tag{8}$$

The dust forcing efficiency $\eta$ is defined as the direct radiative effect per unit optical depth,

$$\eta = \frac{\zeta}{\tau} \tag{9}$$

and in the SW at the surface or TOA are the sensitivity terms in Eqs. 7 and 8, respectively. The atmospheric components of the direct radiative effect and forcing efficiency are interpreted as the differences between their respective TOA and surface values, where uncertainty is obtained by adding the surface and TOA uncertainties in quadrature.

We utilize a Monte Carlo approach to quantify the uncertainty in observational estimates of the SW direct radiative effect and forcing efficiency. Specifically, we repeatedly estimate the terms in Eqs. 4 and 5 adding random error to each of the terms derived from gaussian probability distribution functions having a mean of zero and standard deviations equal to the uncertainties indicated in Table 1. The uncertainties reported here represent the 95% confidence interval about the means.

#### 4.1.2 Observational results

The instantaneous SW dust direct radiative effect and forcing efficiency generated from surface and satellite measurements are shown in Fig. 12. At the surface the SW direct radiative effect is $-15 \pm 1$ W m$^{-2}$, which corresponds to a forcing efficiency





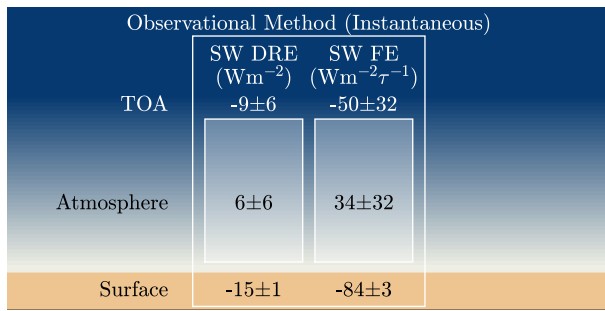

**Figure 12.** Surface and satellite based observational estimates of the clear-sky instantaneous SW direct radiative effect (DRE) and forcing efficiency (FE) at the surface, atmosphere, and TOA. The reported uncertainties represent the estimate's 95% confidence intervals.

of $-84 \pm 3$ W m$^{-2}$ $\tau^{-1}$, and which suggests surface cooling due to scattering and absorption of dust in the atmosphere. At the TOA the SW direct radiative effect is $-9 \pm 6$ W m$^{-2}$ and the accompanying forcing efficiency is $-50 \pm 32$ W m$^{-2}$ $\tau^{-1}$, which represents cooling by scattering of sunlight back out to space by dust. The relatively larger uncertainty in the TOA

forcing values as compared to the surface is related to the number of observations available to calculate each; nearly 2,000 at the surface and just over 40 at TOA. The atmospheric direct radiative effect and forcing efficiency are $6 \pm 6$ W m$^{-2}$ and $34 \pm 32$ W m$^{-2}$ $\tau^{-1}$, representing heating by dust due to absorption from iron oxides in the aerosols (Fig. 6).

### 4.2   Model estimates of the SW and LW instantaneous direct radiative effect and forcing efficiency

We next estimate the dust direct radiative effect and forcing efficiency using output from the RRTM simulations (Sec. 3.1).

#### 4.2.1   Model methodology

The SW and LW dust direct radiative effect is obtained from the differences of the modeled surface or TOA clear and pristine sky fluxes (e.g., Eq. 6). Forcing efficiencies (Eq. 9) are then estimated by regressing $\tau$ onto these direct radiative effect values. We estimate uncertainties again using the Monte Carlo technique. To do so we repeat the calculations of the direct radiative effect and forcing efficiency adding random error to the model fluxes from a gaussian probability distribution function having

mean zero and standard deviation consistent with the errors reported in Fig. 10, and adding random error to $\tau$ based on the instrumental uncertainty (Table 1). We report the 95% confidence intervals for each estimate. The atmospheric components of the direct radiative effect and forcing efficiency, and their associate uncertainties, are obtained in a manner identical to that for the observational estimates.

#### 4.2.2   Model results

The instantaneous dust direct radiative effect and forcing efficiency generated from the radiative transfer model output are shown in Fig. 13. The modeled SW direct radiative effect values are all within the 95% confidence intervals of their respective observational values (Fig. 12, $-16 \pm 1$, $13 \pm 1$, and $-3 \pm 1$ W m$^{-2}$ at the surface, atmosphere, and TOA, respectively). The





| | Instantaneous | | | | | |
|---|---|---|---|---|---|---|
| | RTM DRE (W m$^{-2}$) | | | RTM FE (W m$^{-2}\tau^{-1}$) | | |
| | SW | LW | Net | SW | LW | Net |
| TOA | -3±1 | 1±1 | -2±1 | -17±6 | 4±3 | -13±7 |
| Atmosphere | 13±1 | -2±1 | 12±2 | 76±8 | -8±6 | 68±9 |
| Surface | -16±1 | 3±1 | -14±1 | -93±4 | 12±5 | -81±6 |

**Figure 13.** Shown are estimates of the clear-sky instantaneous SW, LW, and net dust direct radiative effect (DRE, left) and forcing efficiency (FE, right) generated from the radaitive transfer model (RTM) simulations. The reported uncertainties represent the estimate's 95% confidence intervals.

model SW forcing efficiencies at the surface and in the atmosphere are $-93 \pm 4$ and $76 \pm 8$ W m$^{-2}\tau^{-1}$, respectively, and also are in agreement with the observational values. The TOA forcing efficiency is $-17 \pm 6$ W m$^{-2}\tau^{-1}$, agreeing to within 5 W

m$^{-2}\tau^{-1}$ of the observational value. This level of agreement in the model and observation based estimates of the instantaneous SW dust direct radiative effect and forcing efficiency gives confidence in both the derived values and the accuracy of the radiative transfer model.

In the longwave part of the spectrum, at the surface the dust direct radiative effect is $3 \pm 1$ W m$^{-2}$, which corresponds to a forcing efficiency of $12 \pm 5$ W m$^{-2}$ $\tau^{-1}$, and which suggests surface warming due to dust absorptivity. At TOA the LW

direct radiative effect is $1 \pm 1$ W m$^{-2}$ and the accompanying forcing efficiency is $4 \pm 3$ W m$^{-2}$ $\tau^{-1}$, which represents a near balance of warming via absorption and cooling via scattering. The atmospheric component of the direct radiative effect and forcing efficiency are thus $-2 \pm 1$ W m$^{-2}$ and $-8 \pm 6$ W m$^{-2}$ $\tau^{-1}$, which in this case is likely due to the abundance of quartz, feldspar, montmorillonite, and kaolinite (Fig. 6), which have absorption peaks in the atmospheric window region (not shown). Although uncertainty in the radiative transfer model LW output has been assessed via comparison with observed fluxes (Fig.

10), we have less confidence in our estimates of the LW direct radiative effect and forcing efficiency than that for the SW since 1) no credible method exists to independently estimate the LW forcing efficiency from observational data alone, and 2) the relative fractions of quartz and feldspar, which play an important role in determining the LW direct effect, are not constrained by AVIRIS-C.

We estimate the instantaneous net dust direct radiative effect and forcing efficiency by summing the SW and LW components

(Fig. 13). Since the SW values are all greater in magnitude than those in the LW, the net values all have the same sign as that for the SW, implying that during these daytime conditions the SW effect dominates over that for the LW. At the surface, in the atmosphere, and at the TOA the net direct radiative effect is $-14\pm1$, $12\pm2$ and $-2\pm1$ W m$^{-2}$, respectively. The corresponding forcing efficiencies are $-81 \pm 6$, $68 \pm 9$ and $-13 \pm 7$ W m$^{-2}$ $\tau^{-1}$. These dust net forcing estimates imply that surface cooling and atmospheric heating represent the largest perturbations to the radiative budget during dust storms, but that they do not

exactly balance, resulting in a small, but statistically different from zero, cooling at TOA.



### 4.3 Annually averaged dust direct radiative effect

We next estimate an annually and diurnally averaged dust direct radiative effect for clear-sky conditions using the output from RRTM SW and LW based on monthly and 15 minute averaged in situ and reanalysis data. To do so we conducted simulations with RRTM over 24-hour periods corresponding to the fifteenth day of each calendar month, at a 15-minute temporal resolution.

We define the vertical structures of pressure $P(z)$ and specific humidity $q(z)$ by averaging over the profiles collected from the radiosondes launched at the field site during dusty conditions (Tables 1 and A1). We then estimate monthly values for each by scaling those profiles by monthly averages of the field site station surface pressure or GPS retrieved $P_w$, generated for data collected over the 2020–2022 time period. We prescribe the vertical profile of dust extinction based on the ceilometer profiles corresponding to the radiosonde launches, and then scale these by the long-term mean dust optical depth of 0.18. Our data

suggests a relatively small diurnal cycle in this value, and additional simulation with RRTM where we prescribe a dust diurnal cycle based on measurements had little effect on the calculated direct effect and forcing efficiency. We estimate 15 minute averaged soil temperature ($z = 0$ m) retrieved from observations of LW upward flux and 1 m air temperature in a similar fashion from the site meteorological station. We obtain vertical profiles of temperature directly from the JRA-55 reanalysis on the fifteenth of each month averaged over the 2020–2022 time period. Since the lowest height of the reanalysis air temperature

is approximately 60 m AGL, we interpolate temperatures between these heights using the measured 1 m air temperature. We assume constant surface SW and LW albedos. We calculate the annual and diurnally averaged dust direct radiative effect and forcing efficiency directly from the model output. We then assume that the relative uncertainty in these estimates is equivalent to those from the instantaneous values (Fig. 13).

The estimates of the clear-sky annually and diurnally averaged SW, LW, and net (SW + LW) dust direct radiative effect and

forcing efficiency, at the surface and TOA and in the atmosphere, are shown in Fig. 14. The largest discrepancy between the instantaneous and diurnally and annually averaged values is in the SW at the surface, with values that are approximately half those from the instantaneous calculations (Fig. 13), which is due to averaging over the nighttime hours when there is no solar insolation. At the TOA we do not see a similar reduction in the SW direct radiative effect or forcing efficiency because of the non-linear response in the direct radiative effect to changes in the solar zenith angle associated with the particle asymmetry

factor (Fig. 8b); at low solar zenith angle (i.e., the sun is overhead), the direct effect can become positive due to the strong forward scattering of dust. At higher solar zenith angles, when the sun is lower in the sky but the solar insolation is still large, the direct radiative effect is at maximum negative value. The average solar zenith angle corresponding to the instantaneous calculations is 50º, which corresponds to a regime where the direct radiative effect is negative but not at a maximum value, and which happens to be approximately equal to the diurnal average. We note that this non-linearity in the direct effect is far less

pronounced at the surface.

The model results suggest that over land and close to source regions dust cools the climate, as evidenced by the net direct effect value of $-2 \pm 1$ W m$^{-2}$. There is also a net cooling at the surface $-6 \pm 1$ W m$^{-2}$ and warming of the atmosphere $4 \pm 1$ W m$^{-2}$, implying that the presence of dust likely also increases atmospheric stability (Miller et al., 2004). It is important to note that these values correspond to $\tau = 0.18$, which is the average $\tau$ during dust storms. In order to estimate the true long-term



| | Diurnal & Annual Average | | | | | |
| | RTM DRE (W m$^{-2}$) | | | RTM FE (W m$^{-2}\tau^{-1}$) | | |
| | SW | LW | Net | SW | LW | Net |
|---|---|---|---|---|---|---|
| TOA | -3±1 | 1±0.5 | -2±1 | -15±6 | 4±3 | -11±6 |
| Atmosphere | 6±1 | -2±1 | 4±1 | 32±6 | -11±5 | 21±8 |
| Surface | -8±0.4 | 3±1 | -6±1 | -46±2 | 15±5 | -31±5 |

**Figure 14.** Clear-sky annually and diurnally averaged SW, LW, and net direct radiative effect and forcing efficiency of dust at the surface, the TOA, and in the atmosphere, estimated from simulations with a radiative transfer model (RRTM). Uncertainties represent the 95% confidence interval and are based on the uncertainties in the instantaneous values (Fig. 13).

mean direct radiative effect, for example, we would need to scale these values by the fraction of time that dust is present in the atmosphere and the clear-sky fraction. While the latter can be obtained from satellites, it is difficult to reliably estimate the former since we are unable to detect all dust storms occurring during the 2020–2022 time period. However, we speculate that such averaging would likely lead to a uniform reduction in these values by an order of magnitude.

## 5  Comparison to other studies

We next compare our estimates of the direct radiative effect and forcing efficiency with values from other observational and model studies.

### 5.1  Instantaneous SW comparisons

We begin by comparing our estimates of the SW instantaneous forcing efficiency with observational values estimated at the TOA over the Sahara (Kuwano and Evan, 2022), and over Lampedusa at the surface (Di Sarra et al., 2013) and at the surface

and TOA (Di Biagio et al., 2010, study locations in Fig. 15a). Each of these studies report forcing efficiency values averaged over discrete solar zenith angle ranges, and so for these comparisons we recalculated the forcing efficiency estimated from the field site for those same zenith angle ranges.

The surface SW forcing efficiency calculated from our study over the $\mu$ range used in Di Sarra et al. (2013) is $-102 \pm 9$ W m$^{-2}\tau^{-1}$, which is just over half the value of $-177 \pm 17$ W m$^{-2}\tau^{-1}$ reported in their study (Fig. 15b). Similarly, over the $\mu$

range used in Di Biagio et al. (2010), we obtain a forcing efficiency of $-72 \pm 3$ W m$^{-2}\tau^{-1}$, which is less than half the value of $-178 \pm 18$ W m$^{-2}\tau^{-1}$ reported in their study (Fig. 15c). We speculate that there are two main reasons for the discrepancies in the surface SW forcing efficiency estimates. Firstly, the Lampedusa studies included surface albedo values corresponding to both land and water covered surfaces, whereas for this study we limited our study region to footprints that were mostly over land (Fig. 4). As the surface albedo decreases the contrast between clear-sky and dusty scenes increases, resulting in a larger





(in magnitude) forcing efficiency. Secondly, it is also likely that methodological differences also play a role in the discrepancy. Both studies adopted the methods of Satheesh and Ramanathan (2000) to calculate the forcing efficiency, which is equivalent to neglecting the water vapor $P_w$, cosine of the solar zenith angle $\mu$, and albedo $\alpha$ terms in Eq. 4. As such, if $\tau$ is correlated to these terms the resulting forcing efficiency may be biased (Kuwano and Evan, 2022). Furthermore, the criteria for selecting dust in also different; in these two studies dusty scenes are mainly identified via angstrom exponent thresholds. If we calculate

the surface SW forcing efficiency following the methods described in Di Biagio et al. (2010) we obtain a forcing efficiency of $-119 \pm 17$ W m$^{-2}\tau^{-1}$, which is significantly closer to their estimate of $-178 \pm 18$ (not shown). If we also artificially adjust the surface albedo in our calculations to be half the observed value (approximately 0.15 rather than the observed 0.3), then we obtain a SW forcing efficiency of $-145 \pm 21$ W m$^{-2}\tau^{-1}$, which is in agreement with that from Di Biagio et al. (2010). Thus, it is plausible that differences in methodology and environmental characteristics are the main causes of disagreement in the SW

forcing efficiency estimates, rather than differences in the actual radiative properties of the dust.

In order to ensure sufficient data for the remaining comparisons discussed in this section, we use output from RRTM to simulate the SW forcing efficiency at the TOA, which is justified given the agreement between our observation and model based estimates of the SW forcing efficiency (Figs. 12,13). The relative uncertainty in the RRTM output is assumed to be identical to that in Fig. 13, with the absolute uncertainties shown in the subsequent figures. The instantaneous SW TOA forcing

efficiencies from Di Biagio et al. (2010) and Kuwano and Evan (2022) are $-55 \pm 13$ and $-17 \pm 8$ W m$^{-2}\tau^{-1}$, respectively, whereas we obtain a value of $13 \pm 6$ W m$^{-2}\tau^{-1}$ averaged over the solar zenith angle range of roughly 0–45° ($\mu$ from 0.7–1, Fig. 15c). We speculate that, similar to the case for the surface, the disagreement in TOA forcing efficiency is at least in part due to methodological differences. For example, we recalculated the TOA forcing efficiency from Kuwano and Evan (2022) also accounting for variations in surface albedo and $\mu$ (e.g., Eq. 8) obtaining a value that was slightly smaller in magnitude

($-12$ W m$^{-2}\tau^{-1}$) and closer to the value obtained here. Within the atmosphere the forcing efficiency from this study, $85 \pm 7$ W m$^{-2}\tau^{-1}$, and Di Biagio et al. (2010), $129 \pm 23$ W m$^{-2}\tau^{-1}$, are in better agreement that at the surface or TOA, although the magnitudes are still statistically different.

Overall the studies compared here agree that the SW forcing efficiency of dust at the surface is negative, and at the TOA is relatively smaller in magnitude and close to zero, resulting in a positive atmospheric forcing efficiency. These characteristics

are consistent with weakly absorbing aerosols that have minimal TOA radiative effect, and whose main radiative effect is to increase solar absorption in the atmosphere at the expense of the downwelling solar flux at the surface.

## 5.2 Instantaneous LW comparisons

We next compare our instantaneous LW dust forcing efficiency estimates with those from seven other studies. The locations of all studies compared here are shown in Fig. 16a. Similarly to this study, Hansell et al. (2012) used observations and retrievals to

constrain radiative transfer model estimates of the LW direct radiative effect during a 2 week dust storm in Zhangye, China. We then estimated an equivalent forcing efficiency based on their reported mean dust storm optical depth of 0.5. We find statistical agreement between our estimates of the LW instantaneous forcing efficiency, $12 \pm 5$ W m$^{-2}\tau^{-1}$, and that from Hansell et al. (2012), $19 \pm 9$ W m$^{-2}\tau^{-1}$ (Fig. 16b). At the TOA we find that our estimate of $4 \pm 3$ W m$^{-2}\tau^{-1}$ is also in agreement with the





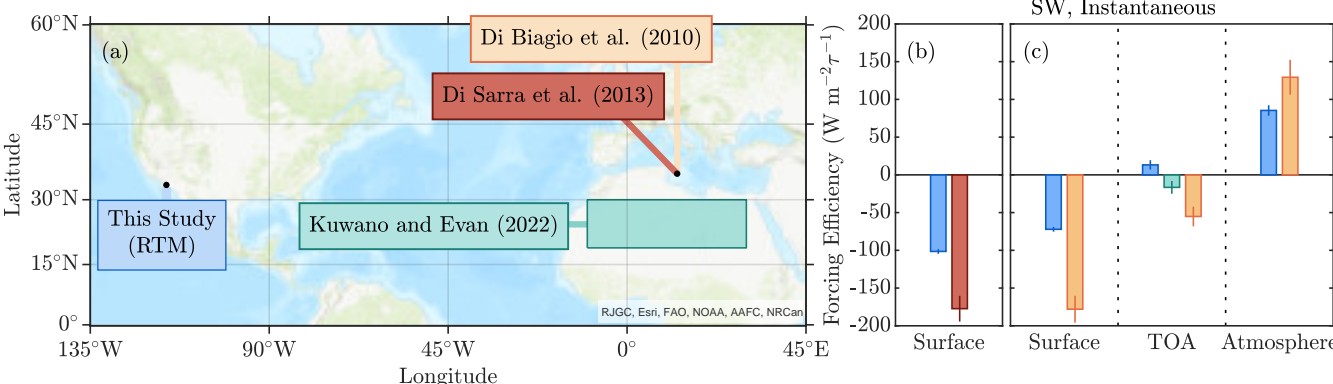

**Figure 15.** (a) Map of the locations or region used to calculate the SW forcing efficiency of dust in this and three other studies (Di Biagio et al., 2010; Di Sarra et al., 2013; Kuwano and Evan, 2022). The corresponding SW forcing efficiency values at the surface, TOA, and in the atmosphere averaged over the $\mu$ intervals (b) 0.34–0.8 and (c) 0.7–1, where colors of the bars are referenced to the colors of the text boxes, and their studies, indicated in (a).

value of $10\pm7$ from Hansell et al. (2012), as well as that from Brindley and Russell (2009), $14\pm10$ W m$^{-2}\tau^{-1}$, corresponding

to the Saada, Morocco location (BR09e in Fig. 16a). Otherwise, the other 13 reported values of the TOA instantaneous LW forcing efficiency are all statistically larger than what we report here, with values ranging from 12 (Song et al., 2022) to 36 (Hsu et al., 2000) W m$^{-2}\tau^{-1}$ (Fig. 16b). The average forcing efficiency from all studies we compare our results to is $20\pm6$ W m$^{-2}\tau^{-1}$, where we note that the uncertainty is poorly constrained since several studies did not report an uncertainty range. We note that the Hansell et al. (2012) TOA and surface values imply an atmospheric LW forcing efficiency of $-5\pm13$ W m$^{-2}\tau^{-1}$,

which agrees with our value of $-8\pm6$ W m$^{-2}\tau^{-1}$.

It is difficult to precisely ascertain the causes of the differences in the LW instantaneous forcing efficiency at TOA. However, the relatively shallow depth of dust over the field site may at least partially explain why our value of the forcing efficiency is smaller than that from these other studies. For example, Hansell et al. (2012) reported an average dust scale height of 3 km during their study, whereas we rarely observed dust layers extending beyond a height of 2 km (e.g., Evan et al., 2023).

### 5.3  Comparison of diurnal and annual averaged SW forcing efficiencies

Lastly, we compare estimates of clear-sky diurnally and annually averaged SW forcing efficiency at the surface, TOA, and atmosphere between this and seven other studies (study locations in Figure 17a). The annually and diurnally averaged SW forcing efficiency at the surface estimated from this study is $-46\pm2$ W m$^{-2}\tau^{-1}$ (Fig. 17b, blue) and is statistically similar to the value of $-48\pm6$ W m$^{-2}\tau^{-1}$ reported by Zhou et al. (2005) for sites with a SW surface albedo of $\sim0.3$ (teal bar and

"o" in Fig. 17). The results from Zhou et al. (2005) corresponding to the sites with surface albedos of approximately 0.2, and the other studies shown in Fig. 17 (Di Biagio et al., 2010; Ge et al., 2010; Valenzuela et al., 2012; García et al., 2014), all report surface forcing efficiencies that are significantly more negative than what we find, having a multi-study mean value of





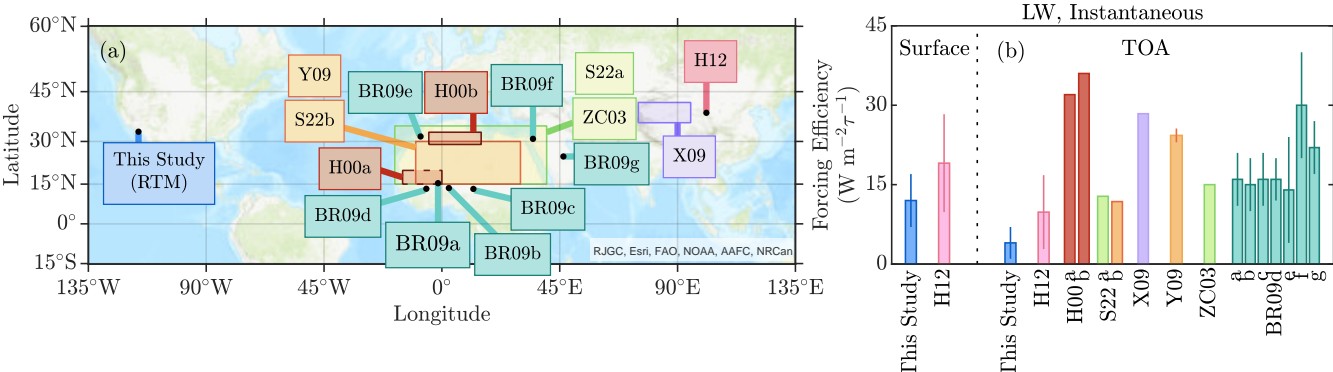

**Figure 16.** (a) Map of the locations or regions used to calculate the LW forcing efficiency of dust in this and seven other studies: Hansell et al. (2012, H12), Hsu et al. (2000, H00a,b), Song et al. (2022, S22a,b), Xia and Zong (2009, X09), Yang et al. (2009, Y09), Zhang and Christopher (2003, ZC03), and Brindley and Russell (2009, BR09a–g). (b) The corresponding LW forcing efficiency values at the surface and TOA, where colors of the bars are referenced to the colors of the text boxes indicated in (a).

$-75 \pm 9$ W m$^{-2}\tau^{-1}$. We speculate that, similar to the case for the instantaneous forcing efficiency (Fig. 15), at least part of this discrepancy is due to methodological differences.

At the TOA our SW forcing efficiency of $-15 \pm 6$ W m$^{-2}\tau^{-1}$ is in agreement with the values of $-26 \pm 11$ and $-16 \pm 9$ W m$^{-2}\tau^{-1}$ from Zhou et al. (2005) and $-17 \pm 7$ W m$^{-2}\tau^{-1}$ from Valenzuela et al. (2012) (Fig. 17b). Similar to the instantaneous case, the TOA forcing efficiency from Di Biagio et al. (2010) of $-46 \pm 6$ W m$^{-2}\tau^{-1}$ is 3-folds larger in magnitude than what we report. For these same four studies we also calculated the atmospheric forcing. Here we find agreement in our estimate of $32 \pm 6$ W m$^{-2}\tau^{-1}$ and those from Zhou et al. (2005) and Di Biagio et al. (2010), which are $36 \pm 14$, $32 \pm 11$, and $32 \pm 7$ W

m$^{-2}\tau^{-1}$, respectively, with the estimate of $57 \pm 14$ W m$^{-2}\tau^{-1}$ from Valenzuela et al. (2012) being significantly larger than that estimated from the other studies.

When averaging across all studies, we obtain an observation based SW forcing efficiency of $-68 \pm 8$, $-24 \pm 8$, and $38 \pm 11$ W m$^{-2}\tau^{-1}$ at the surface, TOA, and in the atmosphere, respectively. Based on these values, the SW direct radiative effect at TOA is $3 \pm 1$ times larger in value than that at the surface, implying that the SW surface cooling by dust is not balanced by

the relatively weaker heating in the atmosphere, resulting in a negative direct effect at TOA. These results also underscore the importance of quantifying the iron oxide content in dust, since these minerals drive solar absorption and thus strongly affect the balances in Fig. 17b (Di Biagio et al., 2019). We are not able to generate an equivalent estimate of the annual and diurnally averaged LW forcing efficiency from the studies represented in Fig. 16 since they do not report these values.

# 6    Summary and conclusions

In this study we used both observations and model output to generate observation and model based estimates of the dust direct radiative effect in the American Desert Southwest. To do so, radiometric and meteorological measurements were obtained over





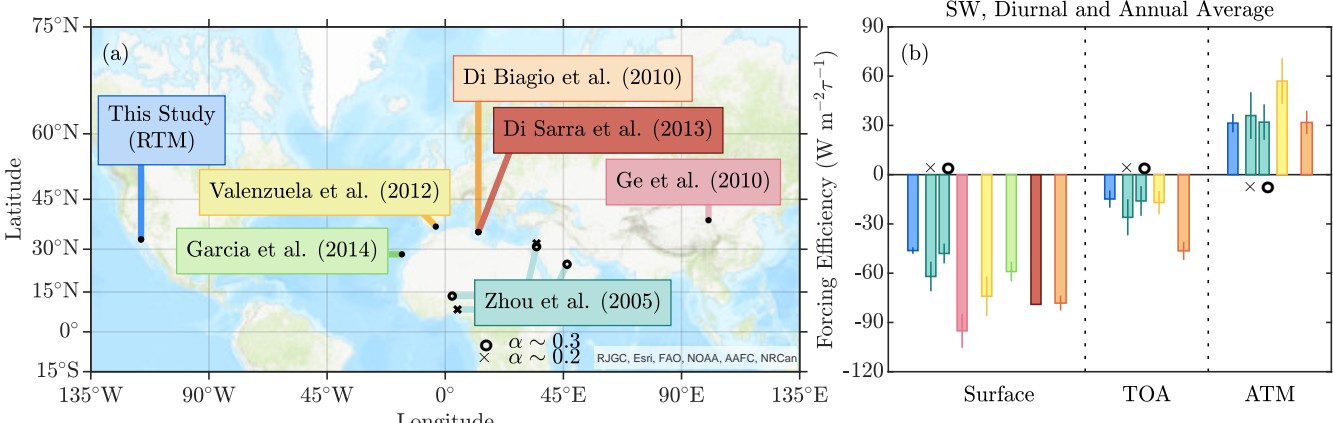

**Figure 17.** (a) A map of the locations or region used to calculate the diurnally and annually averaged SW forcing efficiency of dust for this and six other studies. (b) The corresponding SW forcing efficiency values at the surface, TOA, and in the atmosphere, where colors of the bars are referenced to the colors of the text boxes, and their studies, indicated in (a).

a 3-year period at a field site located in the northwestern Sonoran Desert (Fig. 1). We developed a novel method to estimate the dust SW forcing efficiency and direct radiative effect via our observations alone, in order to generate new estimates of these values at the surface, TOA, and in the atmosphere (Fig. 12). We generated new estimates of the dust refractive index using surface soil mineralogy products from AVIRIS (Figs. 6, 7) and then used these data to model clear-sky fluxes over the field site with RRTM, which we compared to measured fluxes in order to quantify uncertainty in the radiative transfer model (Figs. 9, 10). We then used RRTM to simulate the SW dust direct radiative effect, obtaining agreement between the modeled and observational values (Fig. 13). We also used RRTM to quantify the dust forcing efficiency and direct effect in the LW (Fig. 13). Although the sign of the SW dust direct effect is opposite to that in the LW, since the magnitude of the SW direct effect is 2–3 times larger than that in the LW, we obtained a net direct radiative effect of $-2 \pm 1$ W m$^{-2}$ at TOA that is balanced by a surface cooling of $-6 \pm 1$ W m$^{-2}$ and atmospheric heating of $4 \pm 1$ W m$^{-2}$ (Fig. 14).

We compared our estimates of both the instantaneous and diurnally and annually averaged dust forcing efficiency with values from a number of other studies that estimated these values over a specific site or region using in situ measurements and satellite retrievals. We found that in the SW, our results were more positive than those from three other studies (Fig. 15), but additional analysis suggested that methodological differences between the studies likely played a significant role in the disagreement. We expanded this comparison of the SW forcing efficiency to annually and diurnally averaged values (Fig. 17). Here we found agreement between our estimates and that from several other studies both at the surface, TOA, and in the atmosphere, although we still found that the magnitude of the SW forcing efficiency from our study was smaller than that from all others. We also compared our estimates of the LW instantaneous direct effect with that from other studies, finding that at the surface and at the TOA our LW forcing efficiency was smaller in magnitude than that from these other studies (Fig. 16). Although it is plausible





that methodological differences play a role in differences in the LW forcing efficiencies, the relatively shallow nature of the dust layers advected over the field site (e.g., Fig. 5) relative to that of these other studies is likely a significant underlying cause.

In situ and observation based estimates of the dust direct effect are valuable in terms of understanding how changes in dust concentration affect the radiative energy balance at a local scale. These data are also valuable at the global scale as they provide
an observational constraint on estimates of the direct radiative effect from climate models. However, the apparent sensitivity of the observation based estimates of the SW direct radiative effect to methodology inhibits the utility of these data as a check on model output. As such, and at least in the SW, our results imply that there is a need to adopt a standardized methodology, or at least a standard set of data to be collected, in order to generate data of maximal utility for evaluating climate model output. We suggest that the parameters measured and retrieved in Eqs. 4 and 5 represent a reasonable set of data that, in this
linear framework, account for variability in factors that affect the estimated direct radiative effect. Additionally, a standardized methodology to identify dust from, for example, sun photometers, would also be useful in terms of comparing results among observational studies. Furthermore, it is plausible that a simulator, much like the satellite simulator used to compare cloud cover from climate models and satellites (Klein and Jakob, 1999), could be developed in order to improve the capacity to evaluate model estimates of the dust direct radiative effect.

*Code and data availability.*  Measurements obtained from the CERES SSF level 2 data product can be obtained from https://ceres.larc.nasa. gov/data/. Reanalysis data were acquired from Japan Meteorological Agency (accessed on May 4th, 2023 via https://rda.ucar.edu/datasets/ ds628.0/dataaccess/). AERONET retrievals can be accessed publicly from the AERONET website (https://aeronet.gsfc.nasa.gov). Processed radiosonde data, ceilometer profiles, corrected AERONET aerosol optical depth data, and SW and LW fluxes from the field site are available via the UCSD Library Digital Collection (Evan et al., 2022a). Davis met station data can be publicly accessed from MesoWest station FW7082
Salton City https://mesowest.utah.edu/. $PM_{10}$ data can be accessed from the California Air Resources Board at https://www.arb.ca.gov/ aqmis2/aqdselect.php. For the calibration procedure we utilize the solarPosition calculator (Mikofski, 2016) from the MATLAB file exchange https://www.mathworks.com/matlabcentral/fileexchange/58405-solar-position-calculator (retrieved December 6th, 2022). Dust mineralogy and complex refractive index calculations for this study were done by Blake Walkowiak (Walkowiak, 2022) and scattering properties were obtained from the TAMUdust2020 database (Saito and Yang, 2021) accessed via https://github.com/masasaito/TAMUdust2020. The Rapid
Radiative Transfer Model in the SW (Atmospheric and Environmental Research, 2004) and LW (Atmospheric and Environmental Research, 2010) can be accessed via http://rtweb.aer.com/rrtm_frame.html.

## Appendix A: Radiosonde launches

Shown in Table A1 are the launch dates and times of the radiosondes used to calculate the model fluxes in Fig. 9.





**Table A1.** Days and start times for each sounding considered in RRTM SW and LW.

| Date | Time (UTC) |
| --- | --- |
| 22 February 2020 | 16:59 |
| 22 February 2020 | 18:02 |
| 22 February 2020 | 22:41 |
| 29 February 2020 | 19:23 |
| 29 February 2020 | 21:27 |
| 29 February 2020 | 23:28 |
| 28 February 2021 | 16:05 |
| 28 February 2021 | 18:45 |
| 09 March 2021 | 15:01 |
| 09 March 2021 | 17:59 |
| 09 March 2021 | 19:30 |
| 09 March 2021 | 20:59 |
| 09 March 2021 | 22:30 |
| 09 March 2021 | 23:56 |
| 15 February 2022 | 23:35 |

## Appendix B: Radiometer calibration

As discussed in Sect. 2.1.1, one pyranometer was factory calibrated prior to (2018) and after (2023) acquisition of data for this project, for which the calibration coefficients differed by 1.5% (10.95 and 11.12 $\mu$V W$^{-1}$ m$^2$, respectively). We then calibrated the other instrument by placing the two side-by-side in the upward looking direction for time spans ranging from 1–3 weeks either in La Jolla, CA, or at the field site. We filtered the data consistent with the factory calibration of the reference pyranometer, including only using measurements for solar zenith angle $< 50$ degrees, downward solar flux $> 500$ W m$^{-2}$, and

when the relative difference in the fluxes between the instruments was $< 3\%$. We calculated a cross-calibration coefficient as the slope of the least-squares linear regression of the factory calibrated voltage onto that for the field calibrated instrument, forcing the line through the origin to be consistent with the manufacture's instructions. The resulting cross-calibration coefficients were 1.08 and 1.09 (Fig. B1), all with uncertainties, defined via the 95% confidence interval in the regression slopes, of 0.01%. The resultant calibration coefficients for the field calibrated instrument (factory calibration coefficient multiplied by the cross-

calibration coefficient) were then 10.13, 10.10, 10.10, and 10.24 $\mu$V W$^{-1}$ m$^2$, corresponding to the 2018, 2021, 2022, and 2023 calibration periods, all having $2\sigma$ uncertainties of $\pm 0.001$ $\mu$V W$^{-1}$ m$^2$ (0.01% relative uncertainty).

After cross-calibrating the field calibrated pyranometer (Fig. B1) we noted that the resulting surface SW albedo showed an apparent upward shift after September 1, 2021 (Fig. B2a), which immediately followed a cross-calibration activity. Prior to this date the factory calibrated instrument had been oriented in the downward-facing direction (measuring outgoing radiation).

However, the instruments had been inadvertently reinstalled after this date such that the factory calibrated instrument was





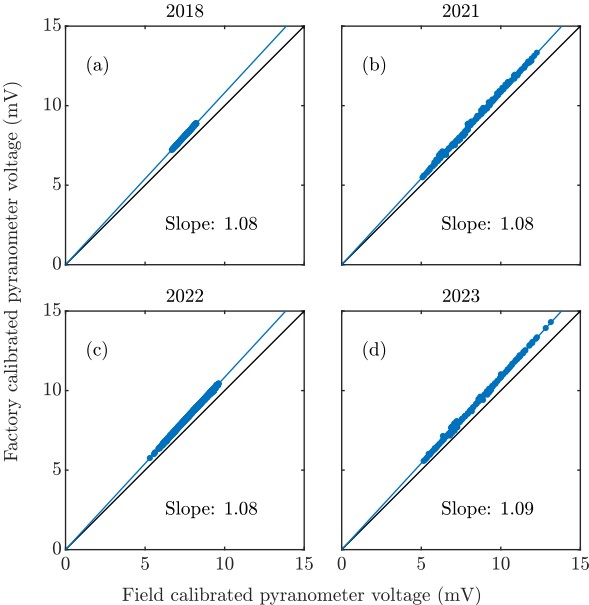

**Figure B1.** Plotted are voltages (blue filled circles) measured from upward looking mounted pyranometers during cross calibration activities in (a,b) La Jolla, CA and at (c,d) the field site. Voltages from the factor calibrated pyranometer are referenced to the vertical axis, and those from the field calibrated instrument are referenced to the horizontal axis. The linear least-squares regression lines are plotted in each panel, and the slope of the lines are also indicated (the 95% uncertainty in each is 0.01%).

oriented in the upward-facing direction (measuring downwelling radiation). To investigate the source of this apparent bias we examined the measured voltages generated by both instruments during 2020 and 2022 during pristine-sky conditions, which were defined as times when the AERONET retrieved aerosol optical depth was less than 0.05.

A plot of uncalibrated voltages from the factory (measured during 2022) and field (measured during 2020) calibrated pyranometers, which is when they were oriented in the upward looking direction, as a function of the cosine of the solar zenith angle $\mu$, shows the expected approximate linear relationship between solar flux and $\mu$ (Fig. B2c). We interpret the ratio of the slopes of the linear least-squares regression lines for these data, both forced through the origin, as the cross-calibration coefficient (factory slope divided by field slope), which in this case is $1.08 \pm 0.003$, in agreement with the cross-calibration coefficients in Fig. B1. We note that the uncertainties in these ratios are determined by summing the relative uncertainties in

the respective regression slopes in quadrature, which are themselves the 95% confidence intervals in the regression slopes. When we repeated this analysis using the uncalibrated voltages when both instruments were oriented in the downward facing direction we obtained a smaller cross-calibration coefficient of $0.92 \pm 0.003$ (Fig. B2d). Applying the different cross-calibration coefficients to the field calibrated pyranometer depending on the instrument orientation (1.08 prior to September 1, 2021 and 0.92 after this date) resulted in a pristine-sky surface solar albedo time series that no longer showed an apparent bias around

this date (Fig. B2b).





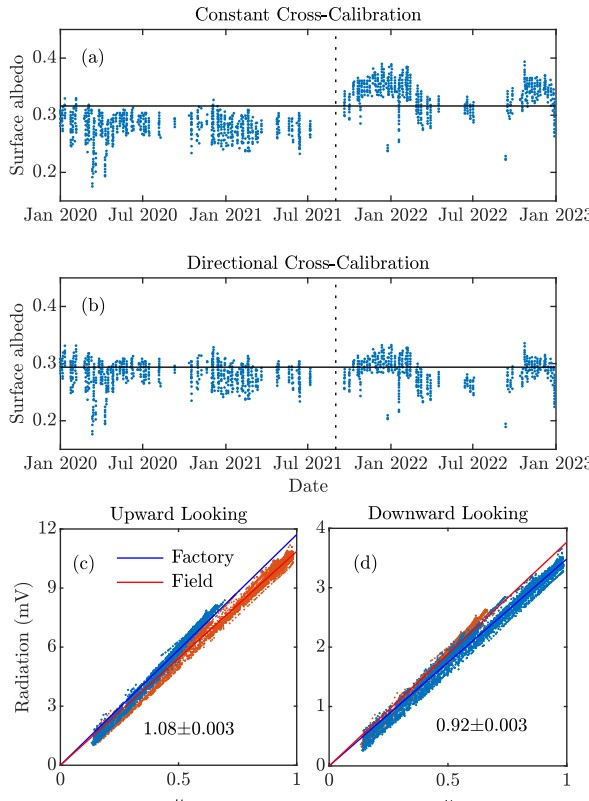

**Figure B2.** (a,b) Time series of surface shortwave albedo generated from voltage measured by upward and downward looking pyranometers at the field site. In (a) a constant cross-calibration coefficient is applied to the field calibrated instrument, and in (b) the calibration coefficient depends on the orientation of the instrument (upward of downward looking). Also shown are plots of measured voltage as a function of the cosine of the solar zenith angle $\mu$ when both instruments are oriented in the (c) upward or (d) downward-looking directions. The specific pyranometer (factory of field calibrated) is indicated in the legend. Also shown in (c,d) are the linear least-squares best fit lines, as well as the ratios of these lines and their respective uncertainties.

Previous work has speculated that pyranometer orientation could impact instrument calibration (Kohsiek et al., 2007), and our results would suggest this is indeed the case, although the cause is not clear. Future work will explore this calibration issue by generating simultaneous measurements from these instruments when both mounted in the upward and then the downward directions. In the meantime, we note that we obtained nearly identical estimates of the dust direct radiative effect and forcing

efficiency when using either cross-calibration method (i.e. a constant value or direction-dependent values). As such, we calibrate the field calibrated instrument by interpolating in time the coefficients derived during the cross-calibration periods when both instruments were mounted in the upward-looking position (Fig. B1).

Turning to the pyrgeometers, one instrument was factory calibrated along with the pyranometer, for which the calibration coefficients differed by 3% (13.11 and 12.72 $\mu$V W$^{-1}$ m$^2$, respectively). We then calibrated the second instrument by placing





the two side-by-side in the upward looking direction similar to as was done for the pyranometers. We filtered the data according to the same criteria used for the factory calibration, including that the measured net flux was $< -40$ W m$^{-2}$, the difference in the instruments temperature were $< \pm 0.5$ C, and that the deviation of the resulting downward longwave flux was $< \pm 5$ W m$^{-2}$. We calculated the cross-calibration coefficient as the slope of the linear least-squares regression of the factory calibrated voltage onto that for the field calibrated instrument, forcing the line through the origin. The resulting cross-calibration coefficients

were 1.09 or 1.10, all with uncertainties, defined via the 95% confidence interval in the regression slopes, of less than 0.02% of the coefficient values (Fig. B3). The resultant calibration coefficients for the field calibrated instrument (factory calibration coefficient multiplied by the cross-calibration coefficient) were then 12.06, 11.97, and 11.63 $\mu$V W$^{-1}$ m$^2$, corresponding to the 2018, 2021, and 2023 calibration periods, all having $2\sigma$ uncertainties of $\pm 0.01$ $\mu$V W$^{-1}$ m$^2$ (0.12% relative uncertainty).

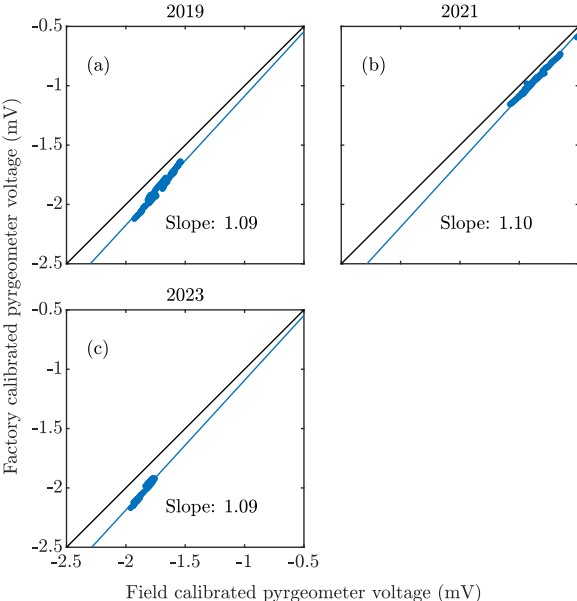

**Figure B3.** Plotted are voltages (blue filled circles) measured from upward looking mounted pyrgeometers during cross calibration activities in (a,b) La Jolla, CA and at (c) the field site. Voltages from the factory calibrated pyrgeometer are referenced to the vertical axis, and those from the field calibrated instrument are referenced to the horizontal axis. The linear least-squares regression lines are plotted in each panel, and the slope of the lines and their 95% confidence intervals are also indicated.

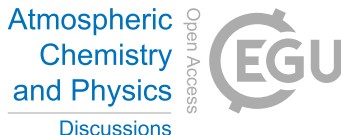

*Author contributions.* AK, AE and RF collected the data. AE and AK designed the experiments and co-wrote the manuscript. AK performed
the numerical experiments and performed the data analysis. BW and AE generated the refractive indices from AVIRIS data. BW and RF
helped edit the manuscript.

*Competing interests.* The authors declare that they have no conflict of interest.

*Acknowledgements.* Funding for this work was provided for by NSF Award AGS-1833173. We thank Tyler Barbero, Trinity Robinson, and
Sophie Wynn for their contributions towards data collection for the project.





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
