# Peer review of "Quantifying the dust direct radiative effect in the Southwestern United States: findings from multiyear measurements"

_Atmospheric Chemistry and Physics, 2024_

## Author Response (AR1)

*Zhibo Zhang, 02 Apr 2024*

Summary

This manuscript presents a study of the direct radiative effect (DRE) of dust aerosols in the southeastern California region using the combined ground and satellite observations, and radiative transfer simulations. In the first step, the instantaneous dust DRE was derived using two methods: 1) the observation based through regression of the measured SW and LW fluxes with respect to measured dust optical thickness; 2) radiative transfer simulation of dust DRE based on observed or derived dust particle properties. In particular, an interesting method is developed to diagnose the dust mineralogy and therefore complex refractive index (CRI) from the AVVRIS-C soil mineralogy retrievals and a dust emission model. After the comparison of the two methods for the instantaneous DRE, the radiative transfer simulations were extended to compute the diurnal DRE. The results are analyzed and discussed in the light of previous studies.

Overall, this is a nice study of the dust DRE based on the comprehensive observations and model simulations, even though it is limited to a specific spot. I enjoy reading it. It fits well with the scope of ACP and is generally well written and clearly presented. On the other hand, I have a few questions and concerns about the methodology used in this study as listed below. I also think some parts of the manuscript need to be clarified and revised significantly before the manuscript can be accepted for publication. Overall, I would recommend a major revision.

Thank you for taking the time to review this manuscript. Your comments and suggestions have been very helpful. We have made most of the changes that you have suggested and think our manuscript has been much improved. We have added a line in the acknowledgements thanking you for your suggestions. Also, as one point of clarification, the observational method is only applied to the SW fluxes, not the SW and LW fluxes, as mentioned in your summary of the study.

- I like the method to diagnose the dust particle mineralogy and refractive index in section 3.1. I think it is interesting, useful and applicable to other measurements like EMIT. But I have a few questions and suggestions.
  - First, there is no discussion about the uncertainty of this method. While I understand that a direct validation is very difficult (perhaps comparing with lab measurements?), I still think there should be an estimation of the uncertainties. For example, I believe AVIRIS soil mineralogy fraction retrievals should have documented uncertainties, right? How does that uncertainty affect the derived dust mineralogy and CRI? It is also interesting and revealing to compare the derived dust CRI with previous studies, especially the database by Di Biagio et al. (2017,2019) For example, the Di Biagio et al. (2019) provides a range of dust SW CRI in their Figure 8. How does the dust CRI in this study (Figure 7) compare to the range of their results?

    This is a fair comment. We have included the uncertainty in the AVIRIS retrieved mineralogy in the discussion of the soil mineralogy (lines 250–252). We then used a monte carlo method to identify the corresponding uncertainty in the complex

refractive index (lines 285–288 and Fig. 7). Since the most radiatively important source of uncertainty was the imaginary part of the refractive index in the SW, which itself stems from the uncertainty in the soil iron oxide content retrievals, we calculated single scatter properties using the mean CRI and the +/- 1-sigma values for the imaginary part of the refractive index (lines 320–325 and Fig. 8). Lastly, we conducted three separate sets of simulations with RRTM, one corresponding to the mean dust single scatter properties, and then one each corresponding to the more and less absorbing (in the SW) cases (lines 342–344 and Figs. 13–17). This effort gave reasonable estimates of the uncertainty in the RRTM output due to the uncertainty in the soil mineralogy retrieval, and led to new insights into the underlying causes of disagreement between the observational and modeled estimates of the DRE in the SW, which are driven by the iron oxide content).

We address the comment regarding the CRI comparison to other studies in the next comment.

- ○ Second, a couple of features of the dust CRI in Figure 7 seem abnormal to me. In the LW, the imaginary part of the CRI has a broad peak between 6 and 10 μm. This absorption peak seems to broad compared to previous studies. For example, most of CRI samples in Di Biagio et al. (2019) has very little absorption at 7-8μm (See Figure 10). But in this study, the imaginary CRI at 7-8 μm is larger than 1. What mineral is responsible for this strong absorption? In addition, the real and imaginary parts of the CRI are not independent but satisfy the the Kramers–Kronig relationship (Bohren and Huffmann, 1983). However, this does not seem to be the case in Figure 7. Can you explain what is the relationship, if any, between the real and imaginary parts of the CRI in your computation?

  We also revised the method used to calculate the CRI from the dust mineral abundances, following the methods described in Song et al (2023), which is described in lines 273–284. This produced CRI estimates that were very similar to those in DiBiaggio 2017 and 2019, and we discuss the characteristics in lines 289–304 and reference the relevant mineral spectra, which are now shown in Fig. C1. We naturally calculated new single scatter properties and conducted new RRTM simulations using these updated (and more reasonable) CRI data. We note that a more thorough comparison of our CRI with other estimates seemed somewhat beyond-the-scope of this paper, particularly since the manuscript is already quite lengthy and since the two DiBiaggio papers provide a comprehensive examination of the vast number of dust CRI estimates used by the community.

- My second major concern is with the linear model to decompose the net SW flux as a linear function of dust aerosol optical depth (DAOD), precipitable water and other factors. I don't understand why a linear model is used here; why not use approximate radiative transfer model

like two-stream approximation? For a dust layer on top of reflecting surface the total reflectance can be approximated as $R_d + \frac{R_sT_d^2}{1-R_dRs}$ where $R_d$ and $T_d$ are the reflectance and transmittance of the dust layer and $R_s$ is surface reflectance. It is not too difficult to add the effect of perceptible water and solar zenith angle to this framework. I suspect that the linear model is simply an approximation when the DAOD is small. In my personal view, I don't think the linear model is intuitive that can shed any light on the underlying physics. I would recommend some discussion based on physical model, e.g., two stream approximation. If linear model is used then there should be some explanation of it theoretical foundation and connection to the radiative transfer.

In this study we develop a linear model of SW radiative fluxes specifically so that we can calculate the dust DRE and FE using only observational data (lines 380–382). In other words, we can calculate the DRE and FE via measurements of tau, net SW fluxes, solar zenith angle, TPW, and surface albedo. We don't need to make any assumptions about the dust CRI, size distribution, particle shape, or vertical distribution. Such an approach is first described by Satheesh and Ramanathan (2000) and is expanded on in Kuwano and Evan (2022). The two-stream model still requires knowledge of the particle single scatter properties and size distribution.

- The discussion of the comparison between the observation-based and model-based DRE results in section 4.2 is not very satisfying or convincing. In my view, the results from the two method are significantly large. For example, the observational based instantaneous DREs at TOA, surface and atmosphere are -9, -15 and 6 W/m^2, respectively. In comparison the results from the model-based computation are -3, -16 and 13 W/m^2. The differences of DRE at TOA and within atmosphere are approaching the limit of uncertainty (I would suggest a statistical test to see the significance of the mean value difference between model based and observation based results). But these significant differences are not really discussed in section 4.2. There is no hypothesis of potential reason or speculation of causes. To me this is a waste of opportunity and seems diminish the purpose of developing two different approaches for computing DRE. I hope to see some mode detailed comparison and insightful analysis of the differences in the revised manuscript.

Because of the incorporation of the AVIRIS mineral uncertainty into the RRTM simulations, we are now able to make what we think is a convincing argument that the differences in the modeled and observational estimates of the SW DRE is actually due to iron oxide content in the AVIRIS soil mineralogy retrievals. Specifically, we proposed that the AVIRIS-retrieved iron oxide content is too large, leading to too strong absorption by dust in the RRTM simulations. We reached out to the AVIRIS/EMIT retrieval team, and they indicated that they also found this to be the case (at the global level) and that now they are processing EMIT with an updated retrieval algorithm that reduces the retrieved soil iron oxide content. This discussion can be found in lines 442–454, is referenced in the abstract and the conclusions, and is consistent with previous work on the importance of iron oxides wrt the SW dust DRE (e.g., Li et al., 2021).

- My last major concern is on the computation of the LW DRE. Based on the information provided in section 3.1.1, the scattering properties of dust, in particular extinction (Figure 8c) in the LW is extrapolated from SW to LW based on the SW DAOD and assumed dust particle

size. As shown in a few previous studies, such extrapolation is highly sensitive to assumption of dust particle size (e.g., see Figure 7 of Song et al. 2018 and Figure 13 of Zheng et al. 2022). This uncertainty should be clearly pointed out and if possible quantify. For example, in addition to the dust PSD used in the current study, the authors can use a coarser dust PSD, for example, FENNEC-SAL PSD from Ryder et al. (2013) to test the sensitivity of LW DRE to dust particle size assumption.

We use the emitted dust PSD from Meng et al. (2022), which does reflect large dust particles and was shown to be consistent with measurements from Claire. We did note this in the initial manuscript, but now we have included a more thorough description of how the Meng et al. dust size distribution was used to estimate the dust single scatter properties, e.g., tau as a function of lambda (lines 311–318).

- In the observation based method, "The atmospheric components of the direct radiative effect and forcing efficiency are interpreted as the differences between their respective TOA and surface values". Note that this method is only method under the 1-D radiative transfer assumption where for a atmosphere column Absorption=1-Reflection-Tansmission. In reality, i.e., 3-D radiative transfer there should be an additional term H that corresponds to the net horizontal flux transfer. For homogeneous dust layer and surface, this 1-D radiative transfer assumption is probably fine but it could face larger error when dust plume is inhomogeneous or there is significant surface inhomogeneity. Nevertheless, this uncertainty should be pointed out and discussed in the revised manuscript.

We have added a statement clarifying that we are assuming horizontally homogeneous conditions, which is an inherent limitation of the approach (lines 418–419).

- "$fs$" in Eq 2 should be $f_s$ with subscript.

We have added an underscore for this symbol in Eq. 2.

Citation: https://doi.org/10.5194/acp-2024-1-RC2

*Anonymous Referee #1, 26 Mar 2024*

General Comments

The authors present an analysis of the radiative effects of dust from the Salton Sea area of southern California, making use of a complementary set of surface observations (radiometric and meteorological), satellite radiation retrievals, and aircraft retrievals of surface soil mineralogy, in conjunction with radiative transfer simulations. The authors make further comparisons of their dust radiative forcing estimates with estimates by other authors for dust over other parts of the globe, including North Africa, the Mediterranean, the Arabian Peninsula, and China. It is an extensive and thorough analysis of the multiple datasets and simulations that have been used, and I have only minor comments.

Thank you for taking the time to review this manuscript. Your comments and suggestions have been very helpful. We have made most, if not all, the changes that you have suggested and think our manuscript has been much improved. We have added a line in the acknowledgements thanking you for your suggestions.

One general point to make is that many of the measurements are daytime-only, and that the simulations during the nighttime period, while these are considered in Sections 4.3 and 5.3 using in-situ and reanalysis data, are relatively less constrained than during the daytime period. AERONET data are not available during the night, nor would the Roundshot camera images be available either, thereby reducing the capability to identify and measure dust storms during this period of the diurnal cycle. Is there a risk of this biasing the results towards the daytime period?

You are correct that the nighttime estimates are poorly constrained. We have altered the manuscript to be clear that the instantaneous DRE/FE estimates are for daytime only (changes made throughout the manuscript), and we have added text (lines 506-511) to indicate that the diurnally averaged values are not constrained by observations, in that we assume a constant tau (lines 506–511 of the revised manuscript).

Something that seems a bit missing in Sections 2.4 and 4.1.2 (for example) is an impression both of the average aerosol optical depth and also its distribution (e.g. its maximum value) over the Salton Sea area. I see that the average value is briefly mentioned much later on in Section 4.3. Quantifying the AOD distribution would help to put the dust loadings over the Salton Sea into context, compared to other dust regions such as the Sahara where the dust loadings may often be more substantial. I see that this possibility is referred to in the middle paragraph of the conclusions.

We have included the mean, standard deviation and maximum dust storm optical depth at the end of Section 2.4 (lines 217–218) of the revised manuscript.

Specific comments

Figures 12-14: in essence these figures are just tables. However I am not going to criticize this, since actually the surface and atmosphere background colors help to provide a more intuitive understanding of what the DRE values indicate.

We agree with this assessment.

Lines 478-480: what are the specific μ ranges? I presume they are similar to the 0-45° range specified on line 501, generally indicating periods during the middle-of-the-day.

The specific μ ranges from Di Sarra and DiBiaggio are now noted in the paragraph beginning on line 521 of the revised manuscript.

Table A1: it may be worth mentioning in the caption what the UTC time is in relation to the local time. We have included the relation of the local/PT time to UTC in the table caption.

Citation: https://doi.org/10.5194/acp-2024-1-RC1